# Task-Driven Subspace Decomposition for Knowledge Sharing and Isolation in LoRA-based Continual Learning

Lingfeng He [1]   De Cheng[✉] [1]   Huaijie Wang [2]   Xi Yang [1]   Nannan Wang [3]   Xinbo Gao [2]

## Abstract

Continual Learning (CL) requires models to sequentially adapt to new tasks without forgetting old knowledge. Recently, Low-Rank Adaptation (LoRA), a representative Parameter-Efficient Fine-Tuning (PEFT) method, has gained increasing attention in CL. Several LoRA-based CL methods reduce interference across tasks by separating their update spaces, typically building the new space from the estimated null space of past tasks. However, they (i) overlook task-shared directions, which suppresses knowledge transfer, and (ii) fail to capture truly effective task-specific directions since these "null bases" of old tasks can remain nearly inactive for new task under correlated tasks. To address this, we study LoRA learning capability from a projection energy perspective, and propose Low-rank Decomposition and Adaptation (LoDA). It performs a task-driven decomposition to build general and truly task-specific LoRA subspaces by solving two energy-based objectives, decoupling directions for knowledge sharing and isolation. LoDA fixes LoRA down-projections on two subspaces and learns robust up-projections via a Gradient-Aligned Optimization (GAO) approach. After each task, before integrating the LoRA updates into the backbone, LoDA derives a closed-form recalibration for the general update, approximating a feature-level joint optimum along this task-shared direction. Experiments indicate that LoDA outperforms existing CL methods.

[1]State Key Laboratory of Integrated Services Networks, School of Telecommunications Engineering, Xidian University, Xi'an, China [2]School of Electronic Engineering, Xidian University, Xi'an, China [3]**AUTHORERR: Missing \icmlaffiliation.** . Correspondence to: De Cheng <dcheng@xidian.edu.cn>.

*Proceedings of the $43^{rd}$ International Conference on Machine Learning*, Seoul, South Korea. PMLR 306, 2026. Copyright 2026 by the author(s).

## 1. Introduction

Continual Learning (CL) studies how a model can continuously acquire new knowledge from a stream of tasks while retaining previously learned capabilities (Kirkpatrick et al., 2017; Qiu et al., 2025). A key challenge is the stability-plasticity dilemma (Kim & Han, 2023): the model must remain plastic enough to adapt to new tasks while keeping stable to avoid forgetting of learned knowledge. With the rise of Pre-Trained Models (PTMs) (Radford et al., 2021; Cheng et al., 2024; Xu et al., 2026), CL has shifted from training from scratch (Kirkpatrick et al., 2017; Wang et al., 2021) to sequentially adapting strong pre-learned representations. Recent open-world recognition studies further emphasize the need to adapt PTMs to emerging categories and concepts (Tang et al., 2025a;b; Tang et al.) in real world. In this regime, Parameter-Efficient Fine-Tuning (PEFT)-based methods (Wang et al., 2022b; Smith et al., 2023; Tan et al., 2024; Xu et al., 2025; Cheng et al., 2026; Tian et al., 2025) introduce lightweight trainable components for adaptation while preserving the generalization of the backbone, achieving remarkable CL performances.

Among various PEFT paradigms, Low-Rank Adaptation (LoRA) (Hu et al., 2022) emerged as a representative and widely-used approach due to its simplicity. It parameterizes the updates with two trainable low-rank matrices while keeping the pre-trained weights frozen. Several works (Zhu et al., 2025; Liang & Li, 2024) have explored LoRA for CL by explicitly isolating task-specific LoRA update subspaces to reduce cross-task interference and mitigate forgetting. However, they face two limitations. Firstly, they mainly focus on subspace isolation, which can discard a large portion of general and transferable directions across tasks, thereby suppressing inter-task knowledge sharing. Secondly, they build the "isolated" subspace for the new task by seeking directions that are inactive on old tasks, typically from the estimated null space of old tasks (Liang & Li, 2024; Wang et al., 2023). But under correlated task distributions in realistic CL, the null spaces of past and new tasks can exhibit substantial overlap, meaning that the estimated isolated bases may still be nearly inactivated by the new task. As a result, they form a "safe zone" rather than a truly effective task-specific subspace. These limitations raise a key ques-

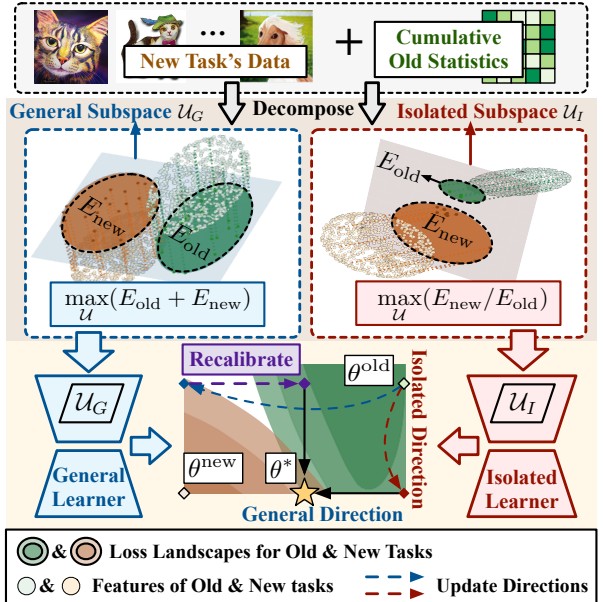

**Figure 1.** LoDA decomposes the update space into general & isolated subspaces ($\mathcal{U}_G$ & $\mathcal{U}_I$) to fix LoRA down-projections. By updating in two subspaces with a post-hoc recalibration, it reaches $\theta^*$ in a shared low-loss region of old and new tasks. $E_{\text{old}}$ and $E_{\text{new}}$ denote the **subspace projection energy** of old and new features. Darker colors in the loss contour map denote lower values.

tion: *how can we properly set LoRA subspaces to preserve transferable directions while learning truly task-specific knowledge, achieving a better stability–plasticity trade-off?*

To address this, we propose a Low-rank Decomposition and Adaptation (LoDA) framework for CL, which performs a task-driven decomposition and derives two LoRA subspaces for knowledge sharing and task isolation. We first reveal that *the impact of a LoRA update on the loss is governed by the projection energy of the task features onto the down-projection's row subspace.* This motivates a data-driven design to freeze down-projections as a gate that selects learnable new feature components while controlling interference with past tasks. Guided by this, LoDA decomposes the update space into two down-projection subspaces using new data and cumulative old data statistics, as shown in Fig.1: (i) a **general subspace** $\mathcal{U}_G$ that yields high projection energy across both old and new tasks ($\mathcal{U}_G = \arg\max_{\mathcal{U}}(E_{\text{old}} + E_{\text{new}})$), capturing directions salient across all tasks and enabling knowledge transfer; (ii) an **isolated subspace** $\mathcal{U}_I$ that exhibit largest new-to-old relative energy ($\mathcal{U}_I = \arg\max_{\mathcal{U}}(E_{\text{new}}/E_{\text{old}})$), identifying updates effective for the new task with little impact on past tasks.

Based on the decomposition, LoDA builds a dual-branch LoRA module: each branch's down-projection is fixed on decomposed subspace bases and the up-projection is learnable. To steer the learning of up-projections toward robust

and cross-class conflict-free directions, we design Gradient-Aligned Optimization (GAO) that encourages its gradient consistency across label-disjoint subsets. After each task, LoDA integrate the decoupled updates into the backbone: (i) Since the general update inevitably induces feature drift of old tasks when learning the new task, we derive a closed-form rescaling matrix to recalibrate it by minimizing feature optimization errors on all tasks to approximate a joint optimum; (ii) Since the isolated update incurs little interference on past tasks, we directly merge it into the backbone.

Our main contributions can be summarized as follows:

- We propose a task-driven decomposition that constructs general and truly isolated down-projection subspaces based on the feature projection energy, decoupling directions for knowledge sharing and isolation.

- We propose LoDA[1], a dual-branch LoRA framework that fixes down-projections on task-driven bases and learns robust up-projections via GAO, with a post-hoc closed-form recalibration for the general branch.

- Experiments demonstrate that LoDA outperforms existing PEFT-based CL methods on multiple benchmarks.

## 2. Related Works

**PEFT-based Continual Learning.** PEFT-based CL methods (Wang et al., 2025a; Chen et al., 2026; He et al., 2026) typically keep a pre-trained model frozen and insert lightweight learnable modules for continual adaptation. Prompt pool-based methods (Wang et al., 2022b; Smith et al., 2023; Gao et al., 2024) build a set of learnable prompts and update relevant prompts during each session to encode task-specific knowledge. PGP (Qiao et al., 2023) and VPT-NSP (Lu et al., 2024) further theoretically deduce anti-forgetting conditions for visual prompts and constrain their updates via gradient projection. Adapter-based methods (Li et al., 2025a; Tan et al., 2024; He et al., 2025; Wang et al., 2026) insert lightweight adapters into transformer blocks and mitigate forgetting via network expansion or feature drift compensation.

Recently, LoRA-based approaches gain increasing attention. SD-LoRA (Wu et al., 2025b) separates the learning of directions and magnitudes to update parameters within a low-loss path. Some methods design regularization schemes for LoRA to mitigate forgetting. O-LoRA (Wang et al., 2023) and InfLoRA (Liang & Li, 2024) constrain LoRA updates to estimated null space of previous tasks, thereby minimizing past output change when learning new tasks. BiLoRA (Zhu et al., 2025) and PLAN (Wang et al., 2025b) impose fixed orthogonal bases on LoRA down-projections, restricting

---

[1]https://github.com/HHHLF/LoDA_ICML2026

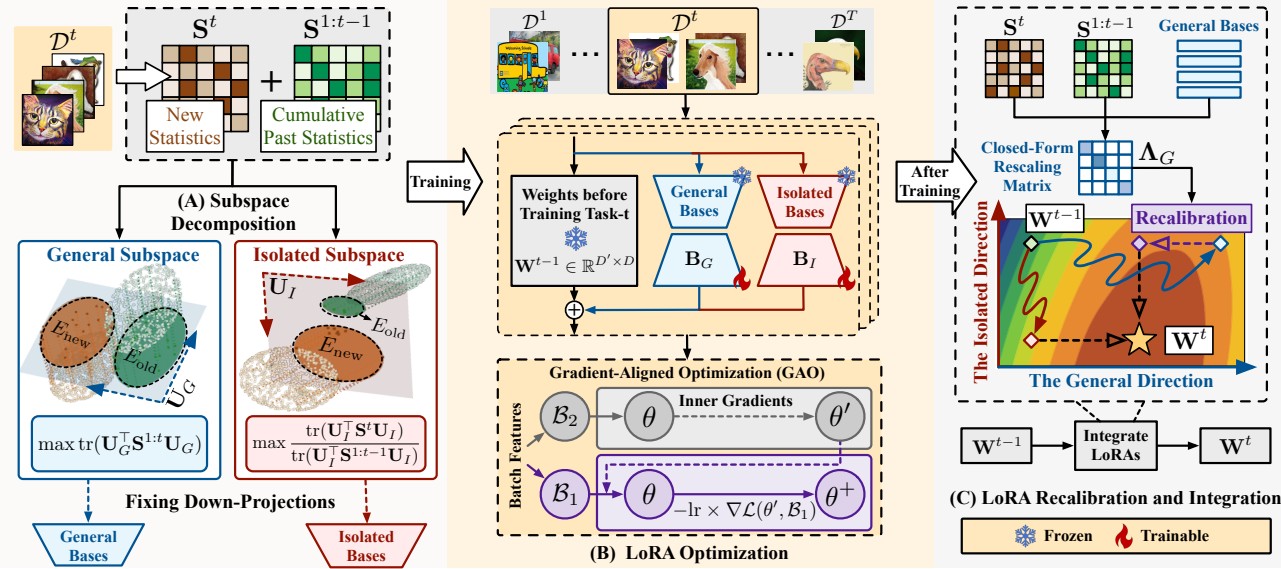

*Figure 2.* Overall framework. At Task-$t$, we freeze backbone weight $\mathbf{W}^{t-1}$ and insert a dual-branch LoRA module. (A) We decompose the update space into general & isolated subspaces for LoRA down-projections. (B) Then we freeze down-projections and train up-projections on $\mathcal{D}^t$ via GAO. (C) After training, the LoRA updates are integrated into backbone with a recalibration on the general branch.

the updates to pre-defined subspace and reducing cross-task interference. However, such orthogonal constraints may fail to uncover truly isolated subspaces while discarding task-shared directions, limiting stability-plasticity trade-off.

**Model Merging.** Model merging (Ilharco et al., 2023; Ortiz-Jimenez et al., 2023) aims to integrate weights trained on different tasks into a single network. Some works (Stoica et al., 2025; Gargiulo et al., 2025) factorize task-wise updates in singular directions and combine them to reduce task interference. Several works have extended this idea to CL. Connector (Lin et al., 2022) designs a convex combination for stability- and plasticity-oriented weights. CoMA (Marouf et al., 2024) designs an EMA-style merging for old and new weights and enhances it with Fisher matrix. BECAME (Li et al., 2025b) models loss changes induced by merging and derives optimal merging coefficients. They mainly rely on local linearity assumptions and gradient-based estimates, leading to approximation errors. We design a feature-level objective and deduce an exact optimum, avoiding such error.

## 3. Methodology

**Problem Definition.** In CL, there are $T$ sequential tasks with non-overlapping classes. The datasets are defined as $\{\mathcal{D}^1, \cdots, \mathcal{D}^t, \cdots, \mathcal{D}^T\}$. $\mathcal{D}^t = \{\mathbf{x}_i^t, y_i^t\}_{i=1}^{N^t}$ denotes the $t$-th task's dataset, where $\mathbf{x}_i^t \in \mathbb{R}^{H \times W \times C}$ is the $i$-th input and $y_i^t \in \mathcal{Y}^t$ is the its class label. $\mathcal{Y}^t$ denotes the label space of the $t$-th task, and $\mathcal{Y}^t \cap \mathcal{Y}^{t'} = \emptyset$ for $t \neq t'$. The objective of CL is to train a model $f(\theta, \cdot)$ sequentially on $T$ tasks and performs well on all seen classes $\mathcal{Y}^1 \cup \cdots \cup \mathcal{Y}^T$.

**Overall framework**. An overview of the proposed method is shown in Fig. 2. Following existing PEFT-based methods (Wang et al., 2022b; Tan et al., 2024), we use a pre-trained Vision Transformer (ViT) (Dosovitskiy et al., 2021) as backbone. At the $t$-th task, for each ViT layer, we freeze the backbone weights $\mathbf{W}^{t-1}$ and attach a dual-branch LoRA module, including a general branch for knowledge sharing and an isolated branch to learn task-specific updates. Guided by our gradient analysis in Sec. 3.1, a task-driven subspace decomposition (Fig. 2(A)) is proposed to derive the general and isolated subspace bases via two projection-energy-based objectives. We freeze each branch's down-projection on the decomposed subspace bases, and then learn robust up-projection on $\mathcal{D}^t$ via a Gradient-Aligned Optimization (GAO) approach, which encourages gradient consistency across label-disjoint subsets(Fig. 2(B)). After each task, we integrate the LoRA updates into the backbone weight. The general branch is recalibrated by a closed-form rescaling matrix $\mathbf{\Lambda}_G$ to approximate a feature-level joint optimum, while the isolated branch is directly merged (Fig. 2(C)).

### 3.1. LoRA Analysis when Fixing Down-Projections

Given the backbone weight $\mathbf{W}^{t-1} \in \mathbb{R}^{D' \times D}$ before the $t$-th task, where $D$ and $D'$ are the input and output dimensions. LoRA trains a low-rank update parameterized by two matrices $\mathbf{A} \in \mathbb{R}^{r \times D}$ and $\mathbf{B} \in \mathbb{R}^{D' \times r}$, while keeping $\mathbf{W}^{t-1}$ fixed. For an input $\mathbf{X} \in \mathbb{R}^D$, the layer output is given by:

$$\mathbf{Y} = \mathbf{X}(\mathbf{W}^{t-1} + \mathbf{BA})^\top \in \mathbb{R}^{D'}. \quad (1)$$

**Theorem 3.1.** *Let $\mathcal{L}(\mathbf{Y})$ be a differentiable loss for $\mathbf{Y}$. Fix $\mathbf{A}$ and update only $\mathbf{B}$ by one gradient descent step $\mathbf{B}' =$*

$\mathbf{B} - \eta \frac{\partial \mathcal{L}}{\partial \mathbf{B}}$. *Then the first-order update $\Delta \mathbf{Y}$ of $\mathbf{Y}$ and the loss decrease is gated by the projected energy $E = \|\mathbf{A}\mathbf{X}^\top\|_2^2$:*

$$\Delta \mathbf{Y} \triangleq \mathbf{Y}' - \mathbf{Y} = -\eta \|\mathbf{A}\mathbf{X}^\top\|_2^2 \frac{\partial \mathcal{L}}{\partial \mathbf{Y}},$$

$$\mathcal{L}(\mathbf{Y}') - \mathcal{L}(\mathbf{Y}) = -\eta \|\mathbf{A}\mathbf{X}^\top\|_2^2 \left\|\frac{\partial \mathcal{L}}{\partial \mathbf{Y}}\right\|_2^2 + o(\|\Delta \mathbf{Y}\|). \tag{2}$$

*If the rows of $\mathbf{A}$ are orthonormal ($\mathbf{A}\mathbf{A}^\top = \mathbf{I}_r$ and $\mathbf{I}_r$ is the $r \times r$ identity matrix), $\|\mathbf{A}\mathbf{X}^\top\|_2^2$ is the squared Euclidean norm of the projection of $\mathbf{X}$ onto the row space of $\mathbf{A}$.*

**Discussion.** Theorem 3.1 reveals: (1) Updating $\mathbf{B}$ changes the output $\mathbf{Y}$ exactly along the steepest descent direction $-\frac{\partial \mathcal{L}}{\partial \mathbf{Y}}$; (2) The magnitude of this change is modulated by the projection energy $E = \|\mathbf{A}\mathbf{X}^\top\|_2^2$, which quantifies how much the input feature $\mathbf{X}$ lies in the subspace selected by $\mathbf{A}$. The detailed proof of Theorem 3.1 is in Appx. B.1.

### 3.2. Low-Rank Subspace Decomposition

Theorem 3.1 indicates that the LoRA's learning capacity is determined by the magnitude of $\mathbf{X}$'s component in the subspace spanned by $\mathbf{A}$. This motivates controlling plasticity on new data and interference with past tasks by designing the down-projection $\mathbf{A}$ in a task-driven manner. Following this, we introduce a dual-branch LoRA with complementary roles, consisting of a general branch LoRA$_G$ (matrices $\mathbf{A}_G, \mathbf{B}_G$) and an isolated branch LoRA$_I$ (matrices $\mathbf{A}_I, \mathbf{B}_I$):

$$\mathbf{Y} = \mathbf{X}(\mathbf{W}^{t-1} + w_G \mathbf{B}_G \mathbf{A}_G + \mathbf{B}_I \mathbf{A}_I)^\top, \tag{3}$$

where $w_G$ is a hyper-parameter weighting the contribution of two branches. We fix $\mathbf{A}_G$ and $\mathbf{A}_I$ and build them via a data-driven low-rank decomposition of the current feature space: $\mathbf{A}_G$ spans a *general subspace* $\mathcal{U}_G \triangleq \mathrm{span}(\mathbf{U}_G)$ that aligns with dominant directions accumulated from all tasks to enable knowledge sharing and inter-task transfer, while $\mathbf{A}_I$ spans an *isolated subspace* $\mathcal{U}_I \triangleq \mathrm{span}(\mathbf{U}_I)$ that is strongly activated by the current task but weakly activated by past tasks, isolating task-specific update directions.

**General Subspace Bases.** Let $\mathbf{X}^i \in \mathbb{R}^{(N^i N^p) \times D}$ denote the concatenated input of task $i$, where $N^p$ denote the number of patch tokens per sample. Let $\mathbf{S}^i \triangleq \mathbf{X}^{i\top}\mathbf{X}^i \in \mathbb{R}^{D \times D}$ denote the second-moment statistics, and $\mathbf{S}^{1:t-1} \triangleq \sum_{i=1}^{t-1} \mathbf{S}^i$ denote the accumulated past statistics. To build a rank-$r$ general subspace, we seek $r$ orthonormal bases $\mathbf{U}_G$ such that the overall projection energy $(E_{\mathrm{old}} + E_{\mathrm{new}})$ of both old and new tasks onto $\mathcal{U}_G = \mathrm{span}(\mathbf{U}_G)$ is maximized:

$$\mathbf{U}_G = \arg \max_{\mathbf{U}^\top \mathbf{U} = \mathbf{I}_r} \Big( \underbrace{\|\mathbf{X}^t \mathbf{U}\|_F^2}_{E_{\mathrm{new}}} + \underbrace{\sum_{i=1}^{t-1} \|\mathbf{X}^i \mathbf{U}\|_F^2}_{E_{\mathrm{old}}} \Big)$$

$$= \arg \max_{\mathbf{U}^\top \mathbf{U} = \mathbf{I}_r} \mathrm{tr}\big(\mathbf{U}^\top \mathbf{S}^{1:t-1}\mathbf{U} + \mathbf{U}^\top \mathbf{S}^t \mathbf{U}\big), \tag{4}$$

where $\mathrm{tr}(\cdot)$ denotes the trace. Since the statistics satisfy $\mathbf{S}^{1:t-1} \succeq \mathbf{0}$ and $\mathbf{S}^t \succeq \mathbf{0}$, the solution is given by the top-$r$

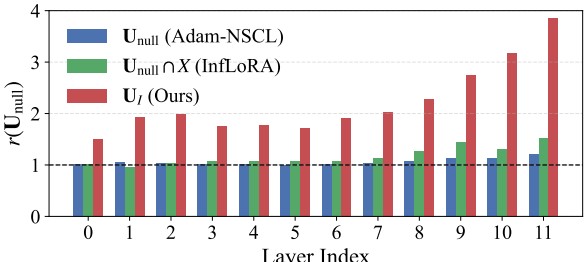

*Figure 3.* Averaged $r^t(\mathbf{U}_{\mathrm{null}})$ in different layers on ImageNetA.

singular vectors of $(\mathbf{S}^{1:t-1} + \mathbf{S}^t)$, where $\mathbf{U}_G = \mathbf{U}_{S[:,1:r]}$ and $[\mathbf{U}_S, \mathbf{\Sigma}_S, \mathbf{V}_S] = SVD(\mathbf{S}^{1:t-1} + \mathbf{S}^t)$. $SVD(\cdot)$ denotes the Singular Value Decomposition (SVD) operator. A brief derivation is provided in Appx.B.2.

**Isolated Subspace Bases.** Aiming to learn task-specific directions, we design an isolated subspace whose directions are strongly driven by task $t$ but weakly driven by past tasks 1:$t-1$. Prior works (Wang et al., 2021; Liang & Li, 2024; Lu et al., 2024) derive the bases by approximating the *null space* $\mathbf{U}_{\mathrm{null}}$ of old tasks, where $\sum_{i=1}^{t-1} \|\mathbf{X}^i \mathbf{U}_{\mathrm{null}}\| \approx 0$. However, real-world task streams can be highly correlated in representation space, and the estimated null space of past tasks can also be nearly inactive for the new task, leading to a small projection $\|\mathbf{X}^t \mathbf{U}_{\mathrm{null}}\|$ and suboptimal subspace isolation. We empirically verify this issue on a challenging yet representative CL benchmark 10S-ImageNetA. We measure how much the new-task inputs project onto the previous-task null space $\mathbf{U}_{\mathrm{null}}$. We define the *relative projection energy* $r^t(\mathbf{U}_{\mathrm{null}})$ of $\mathbf{U}_{\mathrm{null}}$ on task $t$ against all past tasks 1:$t-1$ as:

$$r^t(\mathbf{U}_{\mathrm{null}}) \triangleq \frac{\|\mathbf{X}^t \mathbf{U}_{\mathrm{null}}\|_F^2 / \|\mathbf{X}^t\|_F^2}{\sum_{i=1}^{t-1} \|\mathbf{X}^i \mathbf{U}_{\mathrm{null}}\|_F^2 / (\sum_{i=1}^{t-1} \|\mathbf{X}^i\|_F^2)}. \tag{5}$$

As illustrated in Fig. 3, $r^t(\mathbf{U}_{\mathrm{null}})$ is close to 1.0, indicating that *the estimated null-space directions are activated to a similar extent by both the current and previous tasks*, and thus are not truly effective isolated bases for the new task.

Unlike these methods, we select the $r$-dimensional isolated directions $\mathbf{U}_I \in \mathbb{R}^{D \times r}$ by explicitly maximizing the ratio between (i) the projection energy $E_{\mathrm{new}}$ of task $t$ and (ii) the accumulated energy $E_{\mathrm{old}}$ of all previous tasks $1 : t - 1$:

$$\mathbf{U}_I = \arg \max_{\mathbf{U} \in \mathbb{R}^{D \times r}} \underbrace{\|\mathbf{X}^t \mathbf{U}\|_F^2}_{E_{\mathrm{new}}} / \underbrace{\Big( \sum_{i=1}^{t-1} \|\mathbf{X}^i \mathbf{U}\|_F^2 \Big)}_{E_{\mathrm{old}}}$$

$$= \arg \max_{\mathbf{U} \in \mathbb{R}^{D \times r}} \frac{\mathrm{tr}(\mathbf{U}^\top \mathbf{S}^t \mathbf{U})}{\mathrm{tr}(\mathbf{U}^\top \mathbf{S}^{1:t-1}\mathbf{U})}. \tag{6}$$

**Theorem 3.2** (Computation of $\mathbf{U}_I$). *Since $\mathbf{S}^{1:t-1}$ is the Gram matrix of past features with a huge sample size $\sum_{i=1}^{t-1} N^i N^p \gg D$, we assume it is full rank thus $\mathbf{S}^{1:t-1} \succ \mathbf{0}$. Let $\mathbf{S}^{1:t-1} = \mathbf{L}\mathbf{L}^\top$ be its Cholesky factorization and*

*define* $\tilde{\mathbf{S}}^t \triangleq \mathbf{L}^{-1}\mathbf{S}^t(\mathbf{L}^{-1})^\top$. *Then the optimal* $\mathbf{U}_I$ *of* (6) *is*

$$\mathbf{U}_I = (\mathbf{L}^{-1})^\top \tilde{\mathbf{U}}_I,$$

*where* $\tilde{\mathbf{U}}_I \in \mathbb{R}^{D\times r}$ *consists of* **top-$r$ singular vectors of** $\tilde{\mathbf{S}}^t$.

*Proof.* Using $\mathbf{S}^{1:t-1} = \mathbf{L}\mathbf{L}^\top$, we rewrite the objective as:

$$f(\mathbf{U}) = \frac{\operatorname{tr}\big((\mathbf{L}^\top\mathbf{U})^\top\mathbf{L}^{-1}\mathbf{S}^t(\mathbf{L}^{-1})^\top(\mathbf{L}^\top\mathbf{U})\big)}{\operatorname{tr}\big((\mathbf{L}^\top\mathbf{U})^\top(\mathbf{L}^\top\mathbf{U})\big)}. \quad (7)$$

Define $\tilde{\mathbf{S}}^t \triangleq \mathbf{L}^{-1}\mathbf{S}^t\mathbf{L}^{-\top}$ and make the substitution $\tilde{\mathbf{U}} \triangleq \mathbf{L}^\top\mathbf{U}$. Then Eq.7 can be formulated as follows:

$$f(\mathbf{U}) = \frac{\operatorname{tr}(\tilde{\mathbf{U}}^\top\mathbf{L}^{-1}\mathbf{S}^t(\mathbf{L}^{-1})^\top\tilde{\mathbf{U}})}{\operatorname{tr}(\tilde{\mathbf{U}}^\top\tilde{\mathbf{U}})} = \frac{\operatorname{tr}(\tilde{\mathbf{U}}^\top\tilde{\mathbf{S}}^t\tilde{\mathbf{U}})}{\operatorname{tr}(\tilde{\mathbf{U}}^\top\tilde{\mathbf{U}})}. \quad (8)$$

Since Eq.8 is invariant to the scale of $\tilde{\mathbf{U}}$ and depends only on the directions of $\tilde{\mathbf{U}}$, we impose the normalization $\tilde{\mathbf{U}}^\top\tilde{\mathbf{U}} = \mathbf{I}_r$ without loss of optimality. Then the objective becomes:

$$\max_{\tilde{\mathbf{U}}\in\mathbb{R}^{D\times r}} \operatorname{tr}(\tilde{\mathbf{U}}^\top\tilde{\mathbf{S}}^t\tilde{\mathbf{U}}) \quad \text{s.t.} \quad \tilde{\mathbf{U}}^\top\tilde{\mathbf{U}} = \mathbf{I}_r. \quad (9)$$

The maximum is attained when the columns of $\tilde{\mathbf{U}}$ span the top-$r$ singular vectors of $\tilde{\mathbf{S}}^t$, where $\tilde{\mathbf{U}}_I = \tilde{\mathbf{U}}_{S[:,1:r]}$ and $\left[\tilde{\mathbf{U}}_S, \tilde{\boldsymbol{\Sigma}}_S, \tilde{\mathbf{V}}_S\right] = SVD(\tilde{\mathbf{S}}^t)$ similar to Eq.4. Finally, by substituting back $\tilde{\mathbf{U}} = \mathbf{L}^\top\mathbf{U}$, we derive $\mathbf{U}_I = (\mathbf{L}^{-1})^\top\tilde{\mathbf{U}}_I$. $\square$

### 3.3. LoRA Optimization

**Anchoring Down-Projection Matrices.** We initialize $\mathbf{A}_G$ and $\mathbf{A}_I$ using the derived low-rank subspace bases:

$$\mathbf{A}_G \leftarrow \mathbf{U}_G^\top, \quad \mathbf{A}_I \leftarrow \operatorname{QR}\big(\mathbf{U}_I^\top\big), \quad (10)$$

where $\mathbf{U}_G \in \mathbb{R}^{D\times r}$ and $\mathbf{U}_I \in \mathbb{R}^{D\times r}$ are the bases of the general and isolated subspaces, respectively. $\operatorname{QR}(\cdot)$ denotes the thin-QR factorization that returns the orthonormal factor, so that $\mathbf{A}_I$ has orthonormal rows while preserving $\operatorname{span}(\mathbf{A}_I) = \operatorname{span}(\mathbf{U}_I)$ for stable optimization. $\mathbf{A}_G$ and $\mathbf{A}_I$ are then frozen throughout the optimization on $\mathcal{D}^t$.

**Optimizing Up-Projection Matrices.** After anchoring $\mathbf{A}_G$ and $\mathbf{A}_I$, we further regularize the learnable parameters $\mathbf{B}_G$ and $\mathbf{B}_I$ to induce consistent gradients across diverse classes, so they generalize better and are less susceptible to interference from potential future classes. Specifically, we propose a within-task *Gradient-Aligned Optimization (GAO)* algorithm. Given a batch $\mathcal{B}$, we split it into two label-disjoint subsets $\mathcal{B}_1$ and $\mathcal{B}_2$, GAO implicitly encourages their gradients to align by coupling their gradients. At each step, it first takes a normalized inner update on parameter $\theta$ using the gradient from one subset, and then conducts optimization

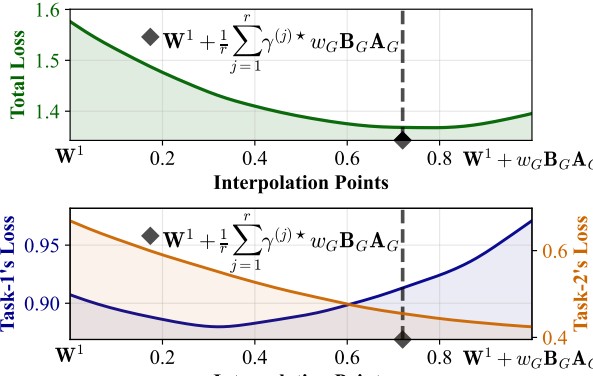

*Figure 4.* Loss values along the general direction (ImageNetA).

on the other subset under the perturbed parameter:

$$\theta^+ \leftarrow \theta - \eta\nabla_\theta\mathcal{L}\left(\theta - \rho\frac{\nabla_\theta\mathcal{L}(\theta,\mathcal{B}_2)}{\|\nabla_\theta\mathcal{L}(\theta,\mathcal{B}_2)\|_2^2}, \mathcal{B}_1\right),$$

$$\theta^{++} \leftarrow \theta^+ - \eta\nabla_\theta\mathcal{L}\left(\theta^+ - \rho\frac{\nabla_\theta\mathcal{L}(\theta^+,\mathcal{B}_1)}{\|\nabla_\theta\mathcal{L}(\theta^+,\mathcal{B}_1)\|_2^2}, \mathcal{B}_2\right),$$

$$(11)$$

where $\rho \sim \mathrm{U}(0, \rho_{\max})$ is a randomized inner-step scale for robustness, and $\eta$ is the learning rate. $\mathcal{L}(\theta,\mathcal{B})$ denotes the standard cross-entropy loss computed on $\mathcal{B}$. Intuitively, it enables learning one group under the gradient interference from another group, suppressing conflicting directions and promoting shared gradients across groups. We provide a theoretical analysis in Appx. B.4 to show that GAO optimizes an explicit gradient alignment objective, while enhancing the anti-interference ability of learned parameters.

### 3.4. LoRA Recalibration and Integration

After training, we recalibrate $\mathrm{LoRA}_G$ via a closed-form matrix that minimizes feature optimization error over tasks $1{:}t$. We then merge updates of two branches into the backbone to reach a joint low-loss region for old and new tasks.

Since the general low-rank branch is designed to capture update directions that are highly activated by both the new task and previous tasks. Therefore, directly adding the optimal update for $\mathcal{D}^t$ leads to uncontrolled representation drifts of past tasks and causes forgetting. To address this, we derive an optimal factor to recalibrate each rank-1 unit in $\mathrm{LoRA}_G$ that minimizes feature-level optimization error. Firstly, we decompose the $r$-rank LoRA $\{\mathbf{B}_G, \mathbf{A}_G\}$ into $r$ rank-1 units $\{\mathbf{B}_G^{(j)}, \mathbf{A}_G^{(j)}\}_{j=1}^r$, where $\mathbf{B}_G\mathbf{A}_G = \sum_{j=1}^r \mathbf{B}_G^{(j)}\mathbf{A}_G^{(j)}$. For each rank-1 unit, we replace its up-projection matrix $\mathbf{B}_G^{(j)}$ by an optimal $\mathbf{B}_G^{(j)\star}$ that achieves the minimal optimization error of both the new task and all previous tasks. The

*Table 1.* Experimental results for 10 incremental tasks on ImageNetR, ImageNetA, CIFAR100, and CUB. We report the averaged results over 3 random seeds. The highest results are in **bold**, and the second highest results are underlined. "FR" denotes the use of feature replay.

| Method | Venue | FR | 10S-ImageNetR | | 10S-ImageNetA | | 10S-CIFAR100 | | 10S-CUB | |
| --- | --- | --- | --- | --- | --- | --- | --- | --- | --- | --- |
| | | | $\mathcal{A}_{Last}\uparrow$ | $\mathcal{A}_{Avg}\uparrow$ | $\mathcal{A}_{Last}\uparrow$ | $\mathcal{A}_{Avg}\uparrow$ | $\mathcal{A}_{Last}\uparrow$ | $\mathcal{A}_{Avg}\uparrow$ | $\mathcal{A}_{Last}\uparrow$ | $\mathcal{A}_{Avg}\uparrow$ |
| L2P (Wang et al., 2022b) | CVPR'22 | ✗ | $72.34_{\pm0.17}$ | $77.36_{\pm0.64}$ | $44.04_{\pm0.93}$ | $51.24_{\pm2.26}$ | $84.06_{\pm0.88}$ | $88.26_{\pm1.34}$ | $67.02_{\pm1.90}$ | $79.62_{\pm1.60}$ |
| DualPrompt (Wang et al., 2022a) | ECCV'22 | ✗ | $69.10_{\pm0.62}$ | $74.28_{\pm0.66}$ | $53.19_{\pm0.74}$ | $64.59_{\pm0.08}$ | $86.93_{\pm0.24}$ | $91.13_{\pm0.32}$ | $68.48_{\pm0.47}$ | $80.59_{\pm1.50}$ |
| CODAPrompt (Smith et al., 2023) | CVPR'23 | ✗ | $73.31_{\pm0.50}$ | $78.47_{\pm0.53}$ | $52.08_{\pm0.12}$ | $63.92_{\pm0.12}$ | $83.21_{\pm3.39}$ | $87.71_{\pm3.17}$ | $77.23_{\pm1.12}$ | $81.90_{\pm0.85}$ |
| InfLoRA (Liang & Li, 2024) | CVPR'24 | ✗ | $74.75_{\pm0.64}$ | $80.67_{\pm0.55}$ | $49.20_{\pm1.12}$ | $60.92_{\pm0.61}$ | $86.75_{\pm0.35}$ | $91.72_{\pm0.15}$ | $70.82_{\pm0.23}$ | $81.39_{\pm0.14}$ |
| SD-LoRA (Wu et al., 2025b) | ICLR'25 | ✗ | $77.34_{\pm0.35}$ | $82.04_{\pm0.24}$ | $55.96_{\pm0.73}$ | $64.95_{\pm1.63}$ | $88.01_{\pm0.31}$ | $92.54_{\pm0.18}$ | $77.48_{\pm0.20}$ | $85.59_{\pm0.44}$ |
| Bi-LoRA (Zhu et al., 2025) | CVPR'25 | ✗ | $77.95_{\pm0.14}$ | $81.52_{\pm0.26}$ | – | – | $87.46_{\pm0.76}$ | $92.50_{\pm0.62}$ | – | – |
| PLAN (Wang et al., 2025b) | ICCV'25 | ✗ | $75.25_{\pm0.42}$ | $80.41_{\pm0.56}$ | – | – | $87.54_{\pm0.31}$ | $92.21_{\pm0.35}$ | – | – |
| LoRA-P&M (Qiu et al., 2025) | NIPS'25 | ✗ | $79.95_{\pm0.18}$ | $85.29_{\pm0.93}$ | $\underline{56.57}_{\pm0.78}$ | $\underline{65.35}_{\pm1.81}$ | $88.45_{\pm0.35}$ | $92.89_{\pm1.13}$ | $\underline{78.29}_{\pm0.50}$ | $\underline{83.39}_{\pm0.61}$ |
| CoSO (Cheng et al., 2025) | NIPS'25 | ✗ | $\underline{81.10}_{\pm0.39}$ | $\underline{85.56}_{\pm0.13}$ | – | – | $\underline{88.77}_{\pm0.16}$ | $\underline{92.99}_{\pm0.23}$ | – | – |
| **LoDA (Ours)** | – | ✗ | $\mathbf{81.93}_{\pm0.20}$ | $\mathbf{86.90}_{\pm0.40}$ | $\mathbf{62.59}_{\pm0.64}$ | $\mathbf{70.87}_{\pm1.61}$ | $\mathbf{90.47}_{\pm0.06}$ | $\mathbf{93.46}_{\pm1.42}$ | $\mathbf{81.74}_{\pm0.78}$ | $\mathbf{89.35}_{\pm0.98}$ |
| SLCA (Zhang et al., 2023) | ICCV'23 | ✓ | $79.35_{\pm0.28}$ | $83.29_{\pm0.46}$ | $61.05_{\pm0.63}$ | $68.88_{\pm2.31}$ | $91.26_{\pm0.37}$ | $94.29_{\pm0.92}$ | $84.68_{\pm0.09}$ | $90.77_{\pm0.79}$ |
| SSIAT (Tan et al., 2024) | CVPR'24 | ✓ | $79.38_{\pm0.59}$ | $83.63_{\pm0.43}$ | $62.43_{\pm1.63}$ | $70.83_{\pm1.63}$ | $91.35_{\pm0.26}$ | $94.35_{\pm0.60}$ | $88.75_{\pm0.38}$ | $93.00_{\pm0.90}$ |
| VQ-Prompt (Jiao et al., 2024) | NIPS'24 | ✓ | $75.68_{\pm0.23}$ | $80.02_{\pm0.18}$ | – | – | $90.27_{\pm0.06}$ | $93.10_{\pm0.84}$ | $86.47_{\pm0.40}$ | $91.37_{\pm0.54}$ |
| DIA (Li et al., 2025a) | CVPR'25 | ✓ | $79.03$ | $85.61$ | $61.69$ | $\underline{71.58}$ | $90.80$ | $94.29$ | $86.73$ | $93.21$ |
| MACIL (Wu et al., 2025a) | ICML'25 | ✓ | $\underline{81.88}_{\pm0.07}$ | $\underline{85.95}_{\pm0.27}$ | $\underline{64.14}_{\pm0.58}$ | $71.45_{\pm1.35}$ | $\underline{91.94}_{\pm0.17}$ | $\underline{94.43}_{\pm0.79}$ | $\underline{90.52}_{\pm0.13}$ | $\underline{93.93}_{\pm0.47}$ |
| **LoDA (Ours) + CA** | – | ✓ | $\mathbf{82.55}_{\pm0.55}$ | $\mathbf{87.18}_{\pm0.42}$ | $\mathbf{66.71}_{\pm0.75}$ | $\mathbf{73.89}_{\pm1.22}$ | $\mathbf{92.15}_{\pm0.21}$ | $\mathbf{94.70}_{\pm1.16}$ | $\mathbf{90.67}_{\pm0.17}$ | $\mathbf{93.97}_{\pm0.56}$ |

*Table 2.* Experimental results for 5 and 20 incremental sessions on ImageNetR, 20 sessions on ImageNetA, and 10 sessions on DomainNet.

| Method | Venue | FR | 5S-ImageNetR | | 20S-ImageNetR | | 20S-ImageNetA | | 10S-DomainNet | |
| --- | --- | --- | --- | --- | --- | --- | --- | --- | --- | --- |
| | | | $\mathcal{A}_{Last}\uparrow$ | $\mathcal{A}_{Avg}\uparrow$ | $\mathcal{A}_{Last}\uparrow$ | $\mathcal{A}_{Avg}\uparrow$ | $\mathcal{A}_{Last}\uparrow$ | $\mathcal{A}_{Avg}\uparrow$ | $\mathcal{A}_{Last}\uparrow$ | $\mathcal{A}_{Avg}\uparrow$ |
| L2P (Wang et al., 2022b) | CVPR'22 | ✗ | $70.83_{\pm0.58}$ | $78.34_{\pm0.47}$ | $69.64_{\pm0.42}$ | $75.28_{\pm0.57}$ | $40.48_{\pm1.78}$ | $49.62_{\pm1.46}$ | $81.17_{\pm0.83}$ | $87.43_{\pm0.95}$ |
| DualPrompt (Wang et al., 2022a) | ECCV'22 | ✗ | $73.05_{\pm0.50}$ | $79.47_{\pm0.40}$ | $66.61_{\pm0.58}$ | $72.45_{\pm0.37}$ | $42.28_{\pm1.94}$ | $53.39_{\pm1.64}$ | $81.70_{\pm0.78}$ | $87.80_{\pm0.99}$ |
| CODAPrompt (Smith et al., 2023) | CVPR'23 | ✗ | $74.91_{\pm0.33}$ | $79.25_{\pm0.53}$ | $69.96_{\pm0.50}$ | $75.34_{\pm0.85}$ | $44.62_{\pm1.92}$ | $54.86_{\pm0.50}$ | $80.04_{\pm0.79}$ | $86.27_{\pm0.82}$ |
| InfLoRA (Liang & Li, 2024) | CVPR-24 | ✗ | $77.52_{\pm0.37}$ | $82.01_{\pm0.12}$ | $71.01_{\pm0.45}$ | $77.28_{\pm0.45}$ | $46.81_{\pm2.30}$ | $57.13_{\pm0.95}$ | $81.45_{\pm0.68}$ | $87.55_{\pm0.57}$ |
| SD-LoRA (Wu et al., 2025b) | ICLR-25 | ✗ | $79.15_{\pm0.20}$ | $83.01_{\pm0.42}$ | $75.26_{\pm0.37}$ | $80.22_{\pm0.72}$ | $49.12_{\pm1.78}$ | $59.63_{\pm1.57}$ | $81.01_{\pm0.42}$ | $86.85_{\pm0.40}$ |
| Bi-LoRA (Zhu et al., 2025) | CVPR-25 | ✗ | $78.32_{\pm0.37}$ | $83.31_{\pm0.52}$ | $72.41_{\pm0.65}$ | $79.28_{\pm0.76}$ | – | – | $76.56_{\pm0.41}$ | $83.20_{\pm0.35}$ |
| LoRA-P&M (Qiu et al., 2025) | NIPS-25 | ✗ | $81.47_{\pm0.56}$ | $85.96_{\pm0.52}$ | $76.37_{\pm0.09}$ | $82.77_{\pm0.31}$ | $\underline{52.27}_{\pm1.84}$ | $\underline{62.15}_{\pm1.33}$ | $\underline{84.35}_{\pm0.48}$ | $\underline{89.43}_{\pm0.27}$ |
| CoSO (Cheng et al., 2025) | NIPS-25 | ✗ | $\underline{82.10}_{\pm0.13}$ | $\underline{86.38}_{\pm0.07}$ | $\underline{78.19}_{\pm0.28}$ | $\underline{83.69}_{\pm0.12}$ | – | – | – | – |
| **LoDA (Ours)** | – | ✗ | $\mathbf{83.37}_{\pm0.14}$ | $\mathbf{87.46}_{\pm0.41}$ | $\mathbf{78.96}_{\pm0.23}$ | $\mathbf{84.94}_{\pm0.17}$ | $\mathbf{55.74}_{\pm1.89}$ | $\mathbf{65.54}_{\pm2.43}$ | $\mathbf{85.49}_{\pm0.11}$ | $\mathbf{90.15}_{\pm0.42}$ |
| SLCA (Zhang et al., 2023) | ICCV-23 | ✓ | $81.01_{\pm0.11}$ | $84.18_{\pm0.28}$ | $74.63_{\pm1.55}$ | $79.92_{\pm1.29}$ | $36.69_{\pm21.31}$ | $56.35_{\pm7.09}$ | $84.35_{\pm0.72}$ | $88.56_{\pm0.21}$ |
| SSIAT (Tan et al., 2024) | CVPR-24 | ✓ | $80.52_{\pm0.07}$ | $84.25_{\pm0.31}$ | $75.67_{\pm0.24}$ | $82.30_{\pm0.36}$ | $59.16_{\pm1.03}$ | $\underline{68.45}_{\pm1.92}$ | $85.11_{\pm0.56}$ | $89.80_{\pm0.34}$ |
| MACIL (Wu et al., 2025a) | ICML-25 | ✓ | $\underline{83.37}_{\pm0.26}$ | $\underline{87.00}_{\pm0.35}$ | $\underline{79.43}_{\pm0.34}$ | $\underline{84.34}_{\pm0.32}$ | $\underline{59.36}_{\pm1.27}$ | $68.03_{\pm2.00}$ | $\underline{86.62}_{\pm0.35}$ | $\underline{90.88}_{\pm0.20}$ |
| **LoDA (Ours) + CA** | – | ✓ | $\mathbf{83.69}_{\pm0.38}$ | $\mathbf{87.45}_{\pm0.42}$ | $\mathbf{81.13}_{\pm0.42}$ | $\mathbf{86.26}_{\pm0.20}$ | $\mathbf{64.47}_{\pm1.27}$ | $\mathbf{72.30}_{\pm1.78}$ | $\mathbf{87.33}_{\pm0.36}$ | $\mathbf{91.40}_{\pm0.17}$ |

objective of $\mathbf{B}_G^{(j)\star}$ can be formulated as follows:

$$
\min_{\mathbf{B}_G^{(j)\star}} \quad \lambda \left\| \mathbf{X}^t (\mathbf{B}_G^{(j)\star}\mathbf{A}_G^{(j)})^\top - \mathbf{X}^t(\mathbf{B}_G^{(j)}\mathbf{A}_G^{(j)})^\top \right\|_F^2
$$
$$
+ \sum_{i=1}^{t-1} \left\| \mathbf{X}^i (\mathbf{B}_G^{(j)\star}\mathbf{A}_G^{(j)})^\top \right\|_F^2 . \quad (12)
$$

The first term matches the LoRA outputs produced by $\mathbf{B}_G^{(j)\star}$ to those produced by the current optimal unit $\mathbf{B}_G^{(j)}$ on the current task $t$. The second term regularizes $\mathbf{B}_G^{(j)\star}$ by minimizing the joint feature optimization error induced by $\mathbf{B}_G^{(j)\star}$ on all previous tasks $1:t-1$. $\lambda$ is a hyper-parameter that balances errors on the new task $\mathcal{D}^t$ and past tasks $\mathcal{D}^{1:t-1}$.

**Theorem 3.3** (Closed-form solution for Eq.12). *Let* $\mathbf{S}^t = \mathbf{X}^{t\top}\mathbf{X}^t \in \mathbb{R}^{D\times D}$ *and* $\mathbf{S}^{1:t} = \sum_{i=1}^{t-1}\mathbf{X}^{i\top}\mathbf{X}^i \in \mathbb{R}^{D\times D}$, *the optimal solution of Eq.12 is given by:*

$$
\mathbf{B}_G^{(j)\star} = \frac{\lambda\mathbf{A}_G^{(j)}\mathbf{S}^t\mathbf{A}_G^{(j)\top}}{\mathbf{A}_G^{(j)}(\lambda\mathbf{S}^t + \mathbf{S}^{1:t-1})\mathbf{A}_G^{(j)\top}}\mathbf{B}_G^{(j)}. \quad (13)
$$

*For each unit, this closed-form solution rescales the optimal weight* $\mathbf{B}_G^{(j)}$ *of the t-th task by a data-dependent factor* $\gamma^{(j)\star}$ *without any approximations, where*

$$
\gamma^{(j)\star} \triangleq \frac{\lambda\mathbf{A}_G^{(j)}\mathbf{S}^t\mathbf{A}_G^{(j)\top}}{\mathbf{A}_G^{(j)}(\lambda\mathbf{S}^t + \mathbf{S}^{1:t-1})\mathbf{A}_G^{(j)\top}} \in [0,1].
$$

A detailed proof is in Appx. B.3.

**Discussion.** We provide a toy example to show knowledge sharing in LoRA$_G$ in Fig.4. We examine the losses along the general update by interpolating from $\mathbf{W}^1$ to $\mathbf{W}^1 + w_G\mathbf{B}_G\mathbf{A}_G$ on 10S-ImageNetA. A mild step along this direction does not harm and can even reduce the loss of Task-1, indicating positive knowledge transfer. However, fully adding Task-2's optimum $w_G\mathbf{B}_G\mathbf{A}_G$ causes feature drifts and notably increases Task-1's loss. This motivates our closed-form recalibration, which deduces optimal rescaling factors $\{\gamma^{(j)\star}\}_{j=1}^r$ to reach a joint low-loss region for both old and new tasks.

For the isolated branch, its updates are highly beneficial for learning the new task, while previous tasks are insensitive to them. Therefore, we directly add them to the weights. The LoRA integration process is formulated as follows:

$$\mathbf{W}^t \leftarrow \mathbf{W}^{t-1} + w_G \mathbf{B}_G \mathbf{\Lambda}_G \mathbf{A}_G + \mathbf{B}_I \mathbf{A}_I, \quad (14)$$

where $\mathbf{\Lambda}_G$ is the diagonal rescaling matrix defined as $\mathbf{\Lambda}_G = \mathrm{diag}\left(\gamma^{(1)\star}, \dots, \gamma^{(j)\star}, \dots, \gamma^{(r)\star}\right) \in \mathbb{R}^{r \times r}$. During inference, we use only the updated backbone weight $\mathbf{W}^t$ and discard all LoRA matrices, yielding an efficient inference stage with no extra parameters. The overall training pipeline is outlined in Algorithm 1 in Appx.A.

## 4. Experiments

**Datasets and Evaluation Metrics.** We evaluate our method on five datasets, namely ImageNetR (Hendrycks et al., 2021a), ImageNetA (Hendrycks et al., 2021b), CIFAR-100 (Krizhevsky et al., 2009), CUB (Wah et al., 2011) and DomainNet (Peng et al., 2019). ImageNet-R contains 30,000 images from 200 categories and diverse styles. ImageNet-A includes 7,500 real-world adversarial examples, spanning 200 classes. CIFAR-100 is a standard CL benchmark, including 60,000 32×32 images from 100 classes. DomainNet consists images from 345 classes across 6 visual domains. Following prior works (Gao et al., 2024), we select the top 200 most populous classes for evaluation.

Following existing PEFT-based methods (Tan et al., 2024; Wu et al., 2025a), we report results using 2 metrics: (1) Last accuracy ($\mathcal{A}_{last}$), defined as the accuracy over all classes after the final session and (2) Average accuracy ($\mathcal{A}_{Avg}$), defined as the mean accuracy across all incremental sessions.

**Implementation Details.** All experiments are conducted on a single NVIDIA RTX 3090 GPU. We use ViT-B/16 (Dosovitskiy et al., 2021) pre-trained on ImageNet-21K (Russakovsky et al., 2015) as the backbone. The model is optimized with an SGD optimizer with a Cosine Annealing scheduler. We train each task for 20 epochs with a batch size of 48. The initial learning rate is set to 0.02 for ImageNetR, 0.005 for CUB, and 0.01 for other datasets. Following (Wu et al., 2025a), we insert LoDA into all attention blocks with rank 32, and employ a cosine classifier for training the classification task. The hyper-parameters $w_G$ in Eq.3 is set to 0.5 and $\lambda$ in Eq.12 is set to 3.0. $\rho_{\max}$ for GAO is set to 0.7 for ImageNetR and 0.3 for other datasets.

**LoDA+CA: an Enhanced Variant with Feature Replay.** In PEFT-based continual learning, performance is often sensitive to the classifier design, particularly when Gaussian sampling-based feature replay is used. To isolate this factor and ensure fair comparison, we implement two protocols within the same framework: (1) **LoDA** follows the standard end-to-end training without any post-hoc classifier adjustment; (2) **LoDA+CA** is an enhanced variant that integrates the widely used Classifier Alignment (CA) technique (Zhang et al., 2023) into the training pipeline.

### 4.1. Comparison with the State-of-the-Arts

We compare LoDA with state-of-the-art PEFT-based approaches, including prompt-based methods (Wang et al., 2022b; Jiao et al., 2024; Wang et al., 2022a), adapter-based methods (Tan et al., 2024; Gao et al., 2023) and LoRA-based methods (Wang et al., 2025b; Wu et al., 2025b; Liang & Li, 2024; Cheng et al., 2025; Qiu et al., 2025) on five CL benchmarks with various settings, as shown in Tab.1 and Tab.2. "FR" denotes the use of feature replay techniques.

The results demonstrate the superiority of our method. Without feature replay, LoDA consistently surpasses the strongest approach CoSO by 0.80%–1.70% in $A_{Last}$. With classifier alignment, LoDA+CA achieves the best performance across all settings, outperforming the feature replay-based SOTA MACIL by 0.15%–5.11% in $A_{Last}$. Notably, the gains are substantial on the more challenging ImageNetA and ImageNetR datasets, where backbone features alone are insufficient for discrimination, but smaller on easier datasets such as CIFAR100 and CUB, where strong pre-trained representations reduce the need for backbone adaptation.

### 4.2. Ablation Studies

We conduct ablation studies on components of the LoDA framework, including the dual-branch LoRA module (*i.e.*, LoRA$_G$ & LoRA$_I$) and the GAO training approach. The results are shown in Tab.3. We also built a naive baseline (Idx-1 in Tab. 3), which sequentially adapts the PTM using the standard single-branch LoRA module. To further analyze its effectiveness in balancing stability and plasticity, we visualize the accuracy curves of old tasks, the new task, and overall performance, as shown in Fig. 5.

**The Effectiveness of the Dual-Branch LoRA.** As shown in Tab. 3, LoRA$_G$ (Idx-2) improves $\mathcal{A}_{Last}$ by 6.49%–8.34% compared to baseline (Idx-1). It restricts adaptation to the task-shared subspace and approximates a feature-level joint optimum, which well balances performances on old and new tasks. LoRA$_I$ (Idx-3) boosts $\mathcal{A}_{Last}$ by 5.99%–7.45% by seeking updates that maximize gains on new task while minimizing interference with past tasks. Notably, enabling both LoRA$_G$ and LoRA$_I$ (Idx-4) further boosts $\mathcal{A}_{Last}$ by 1.38%–2.15% over the best single-branch variant, highlighting their complementary roles. Fig.5 provides further evidence: (i) LoRA$_G$ yields a stable shared update that accounts for performance over all tasks, reinforcing past knowledge; (ii) LoRA$_I$ explores novel directions beyond old feature spaces and merges them intact into the backbone, integrating new knowledge and yielding higher new-task accuracy.

**The Effectiveness of GAO.** Building on the dual-branch

*Table 3.* Ablation Studies on ImageNetR and ImageNetA datasets under 10 and 20 incremental sessions.

| Idx | LoRA$_G$ | LoRA$_I$ | GAO | 10S-ImageNetR | | 10S-ImageNetA | | 20S-ImageNetR | | 20S-ImageNetA | |
|---|---|---|---|---|---|---|---|---|---|---|---|
| | | | | $\mathcal{A}_{Last}\uparrow$ | $\mathcal{A}_{Avg}\uparrow$ | $\mathcal{A}_{Last}\uparrow$ | $\mathcal{A}_{Avg}\uparrow$ | $\mathcal{A}_{Last}\uparrow$ | $\mathcal{A}_{Avg}\uparrow$ | $\mathcal{A}_{Last}\uparrow$ | $\mathcal{A}_{Avg}\uparrow$ |
| 1 | – | – | – | $72.09_{\pm1.02}$ | $81.02_{\pm1.00}$ | $53.24_{\pm1.10}$ | $65.14_{\pm1.41}$ | $68.25_{\pm1.18}$ | $78.34_{\pm0.50}$ | $45.21_{\pm5.51}$ | $60.79_{\pm0.16}$ |
| 2 | ✓ | – | – | $79.19_{\pm0.26}$ | $85.18_{\pm0.94}$ | $59.62_{\pm1.87}$ | $68.84_{\pm2.45}$ | $76.59_{\pm0.39}$ | $83.26_{\pm0.70}$ | $51.70_{\pm0.31}$ | $62.23_{\pm1.74}$ |
| 3 | – | ✓ | – | $79.22_{\pm0.50}$ | $85.22_{\pm0.73}$ | $59.89_{\pm0.68}$ | $68.97_{\pm2.03}$ | $75.70_{\pm0.78}$ | $83.11_{\pm1.08}$ | $51.20_{\pm0.73}$ | $61.97_{\pm0.62}$ |
| 4 | ✓ | ✓ | – | $81.05_{\pm0.22}$ | $86.20_{\pm0.75}$ | $61.27_{\pm1.30}$ | $69.84_{\pm2.29}$ | $78.62_{\pm0.46}$ | $84.47_{\pm0.43}$ | $53.85_{\pm0.79}$ | $63.71_{\pm2.21}$ |
| 5 | ✓ | ✓ | ✓ | $\mathbf{81.93_{\pm0.20}}$ | $\mathbf{86.91_{\pm0.40}}$ | $\mathbf{62.59_{\pm0.64}}$ | $\mathbf{70.87_{\pm1.61}}$ | $\mathbf{78.96_{\pm0.23}}$ | $\mathbf{84.94_{\pm0.17}}$ | $\mathbf{55.74_{\pm1.89}}$ | $\mathbf{65.54_{\pm2.43}}$ |

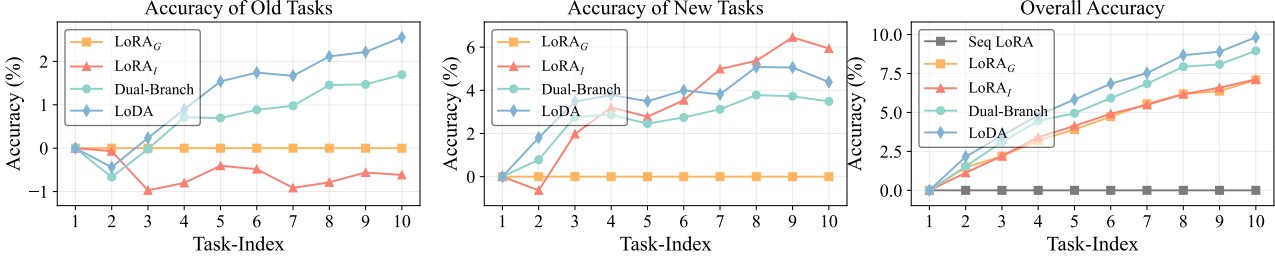

*Figure 5.* Relative accuracy curve of old tasks, new tasks and the overall performances on 10S-ImageNetR.

*Table 4.* Comparison with different subspace isolation methods.

| Dataset | Method | $\mathcal{A}_{Last}\uparrow$ | $\mathcal{A}_{Avg}\uparrow$ |
|---|---|---|---|
| 10S-ImageNetR | Random Orthogonal Bases | $68.23_{\pm0.18}$ | $79.05_{\pm0.62}$ |
| | Adam-NSCL (Wang et al., 2021) | $74.50_{\pm0.77}$ | $82.29_{\pm1.12}$ |
| | InfLoRA (Liang & Li, 2024) | $75.15_{\pm0.37}$ | $82.57_{\pm0.84}$ |
| | **Relative Energy Maximization (Ours)** | $\mathbf{79.22_{\pm0.50}}$ | $\mathbf{85.22_{\pm0.73}}$ |
| 10S-ImageNetA | Random Orthogonal Bases | $53.48_{\pm1.47}$ | $64.34_{\pm1.86}$ |
| | Adam-NSCL (Wang et al., 2021) | $55.95_{\pm1.84}$ | $66.94_{\pm2.03}$ |
| | InfLoRA (Liang & Li, 2024) | $57.10_{\pm0.96}$ | $67.55_{\pm1.94}$ |
| | **Relative Energy Maximization (Ours)** | $\mathbf{59.89_{\pm0.68}}$ | $\mathbf{68.97_{\pm2.03}}$ |

*Table 5.* Comparison of our closed-form rescaling for LoRA$_G$ with existing model merging strategies. "CoMA ($w = \alpha$)" denotes the result when setting the merge coefficient in CoMA to $\alpha$.

| Dataset | Strategy | $\mathcal{A}_{Last}\uparrow$ | $\mathcal{A}_{Avg}\uparrow$ |
|---|---|---|---|
| 10S-ImageNetR | CoMA ($w = 0.1$) (Marouf et al., 2024) | $76.93_{\pm0.65}$ | $81.89_{\pm0.95}$ |
| | CoMA ($w = 0.3$) (Marouf et al., 2024) | $78.39_{\pm0.78}$ | $84.48_{\pm1.01}$ |
| | CoMA ($w = 0.5$) (Marouf et al., 2024) | $77.95_{\pm0.55}$ | $84.30_{\pm0.90}$ |
| | $\theta = \frac{t-1}{t}\theta_{t-1} + \frac{1}{t}\theta_t$ (Lin et al., 2022) | $78.43_{\pm0.20}$ | $84.70_{\pm1.06}$ |
| | **Closed-Form Rescaling ($\lambda = 3.0$)** | $\mathbf{79.19_{\pm0.26}}$ | $\mathbf{85.18_{\pm0.94}}$ |
| | **Closed-Form Rescaling ($\lambda = 10.0$)** | $78.96_{\pm0.14}$ | $85.10_{\pm0.44}$ |
| 10S-ImageNetA | CoMA ($w = 0.1$) (Marouf et al., 2024) | $58.11_{\pm0.97}$ | $66.97_{\pm2.15}$ |
| | CoMA ($w = 0.3$) (Marouf et al., 2024) | $58.84_{\pm0.67}$ | $68.46_{\pm1.98}$ |
| | CoMA ($w = 0.5$) (Marouf et al., 2024) | $57.84_{\pm1.41}$ | $68.01_{\pm1.56}$ |
| | $\theta = \frac{t-1}{t}\theta_{t-1} + \frac{1}{t}\theta_t$ (Lin et al., 2022) | $58.79_{\pm1.11}$ | $68.20_{\pm2.19}$ |
| | **Closed-Form Rescaling ($\lambda = 3.0$)** | $\mathbf{59.62_{\pm1.87}}$ | $\mathbf{68.84_{\pm2.45}}$ |
| | **Closed-Form Rescaling ($\lambda = 10.0$)** | $59.75_{\pm1.21}$ | $68.68_{\pm1.34}$ |

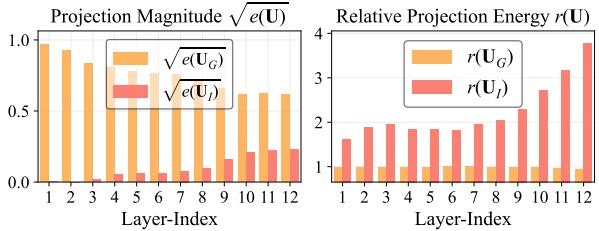

*Figure 6.* Projection magnitude and relative energy across different transformer layers on 10S-ImageNetA.

structure, GAO further improves $\mathcal{A}_{Last}$ by 0.34%–1.89%. As shown in Fig.5, it increases accuracy of both old and new tasks with larger improvements on old tasks. We attribute this to the regularization of gradient consistency, which filters out conflicting gradients early and yields more robust update directions that are less prone to future interference.

### 4.3. In-Depth Analysis

**Comparison with other task isolation methods.** To validate the rationality of constructing the isolated subspace by maximizing the relative projection energy $r(\mathbf{U})$, we compare the performance of LoRA$_I$ (w/o LoRA$_G$) with existing task isolation methods, as shown in Tab.4. Adam-NSCL (Wang et al., 2021) and InfLoRA (Liang & Li, 2024) focus on directions that are weakly activated by past tasks while overlooking their activation on the new task, resulting in sub-optimal performances. In contrast, our objective (Eq.6) enforces a maximal activation contrast between the new and past tasks, deriving truly task-specific subspace.

**Closed-Form Rescaling *v.s.* Existing Merging Strategies.** We compare performance of LoRA$_G$ (w/o LoRA$_I$) using our closed-form rescaling against existing merging strategies, as shown in Tab.5. Our feature-level objective in Eq.12 admits an exact closed-form solution that minimizes feature degradation from both past- and new-task optima, avoiding extra gradient computation (CoMA (Marouf et al., 2024)) and local linearity approximations. Moreover, we assign a unique rescaling factor for each rank-1 unit, which is more informative than direct averaging ($\theta = \frac{t-1}{t}\theta_{t-1} + \frac{1}{t}\theta_t$).

**Visualization and Analysis of the Projection Energy on Two Subspaces.** We visualize normalized projection magnitude defined as $\sqrt{e(\mathbf{U})} = \sqrt{\|\mathbf{X}^t\mathbf{U}\|_F^2/\|\mathbf{X}^t\|_F^2}$ of the

*Table 6.* Computational and storage overhead on 20S-ImageNetR.

| Method | Training Stage | | Inference Stage | | |
|---|---|---|---|---|---|
| | Extra Memory | Training GFLOPs | Additional Params | Inference GFLOPs | $\mathcal{A}_{Last}$ |
| InfLoRA (Liang & Li, 2024) | 11.3MB | 17.67 | 0.00M | 17.60 | 71.01 |
| SD-LoRA (Wu et al., 2025b) | 14.1MB | 20.51 | 0.00M | 17.60 | 75.26 |
| VPT-NSP (Lu et al., 2024) | 28.4MB | 17.72 | 0.09M | 17.72 | 76.13 |
| VPT-CPG (Lu et al., 2025) | 28.4MB | 17.72 | 1.66M | 17.72 | 75.33 |
| **LoDA (Ours)** | 27.0MB | 18.06 | 0.00M | 17.60 | **78.96** |

current task onto the two subspaces, and the new-to-past relative projection energy $r(\mathbf{U})$ defined in Eq.5 on 10S-ImageNetA, as shown in Fig.6. $\mathbf{U}_G$ dominates the projection energy across layers, indicating that most feature components lie in task-shared directions and providing evidence of strong task correlation. $\mathbf{U}_I$ exhibits a larger projection and higher relative energy in deeper layers, suggesting that deeper layers emphasize more task-specific features.

**Overhead Analysis.** We report the computational and storage overhead of LoDA on 20S-ImageNetR in comparison with representative methods, as shown in Tab.6. LoDA incurs a modest storage overhead by maintaining the cumulative statistics $\mathbf{S}^{1:t-1} \in \mathbb{R}^{L \times (D \times D)}$ in $L$ layers, which costs 27.0MB for ViT-B/16 with $L = 12$ and $D = 768$. This cost is *independent of the number of tasks*, thus scaling well to long task streams. LoDA introduces some training overhead (18.06 GFLOPs) due to the dual-branch LoRA, but incurs no extra parameters and overhead for inference (0.00M, 17.60 GFLOPs), while achieving the best performances.

## 5. Conclusion

In this work, we show that LoRA's learning capability is governed by task feature projection onto its down-projection subspace, providing a principled way to control knowledge sharing & isolation in CL. Based on this insight, we propose LoDA, which decomposes the update space into general and truly isolated subspaces via two projection-energy-based objectives. LoDA then builds a dual-branch LoRA module that fixes down-projections on the decomposed bases and learns robust up-projections via Gradient-Aligned Optimization (GAO). After training, LoDA recalibrate the general branch via a closed-form rescaling matrix to approximate a feature-level joint optimum. Overall, LoDA offers a novel view for CL, showing that subspace-aware adaptation provides an effective paradigm for stability-plasticity trade-off.

## Acknowledgements

This work was supported in part by the National Key R&D Program of China under Grant No.2023YFA1008600, in part by the National Natural Science Foundation of China under Grants 62576262, U22A2096, in part by the Key Research and Development Program of Shaanxi Province under grant 2024SF-YBXM-647, in part by the Fundamental Research Funds for the Central Universities under Grant QTZX25083, QTZX23042.

## Impact Statement

This paper presents work whose goal is to advance the field of machine learning. There are many potential societal consequences of our work, none of which we feel must be specifically highlighted here.

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

## A. Training Pipeline

The overall procedure of our algorithm is outlined in Algorithm 1: (1) Before training, we run the frozen backbone $f(\cdot; \mathbf{W}^{t-1})$ on $\mathcal{D}^t$ to cache per-layer inputs and compute statistics $\mathbf{S}^t$. Using $\mathbf{S}^{1:t-1}$ and $\mathbf{S}^t$, we derive the general and isolated subspace bases $\mathbf{U}_G$ and $\mathbf{U}_I$, and initialize the LoRA down-projections $\mathbf{A}_G$ and $\mathbf{A}_I$. (2) During training, we optimize the up-projections $\mathbf{B}_G$ and $\mathbf{B}_I$ on $\mathcal{D}^t$ with the Gradient-Aligned Optimization (GAO) algorithm. (3) After training, we derive a closed-form matrix $\mathbf{\Lambda}_G$ to rescale each rank-1 unit in $\text{LoRA}_G$ , and merge both branches into the backbone. Finally, we update the cumulative statistics $\mathbf{S}^{1:t-1}$ by adding the current-task statistics $\mathbf{S}^t$.

---

**Algorithm 1:** Training pipeline of the proposed Low-rank Decomposition and Adaptation (LoDA).

---

**Input:** Task sequence $\{\mathcal{D}^1, \ldots, \mathcal{D}^T\}$, where $\mathcal{D}^t = \{\mathbf{x}_i^t, y_i^t\}_{i=1}^{N^t}$; ViT model $f(\cdot)$ with weight $\mathbf{W}^0$, two LoRA branches
$\quad\quad$ $\text{LoRA}_G = \{\mathbf{A}_G, \mathbf{B}_G\}$ and $\text{LoRA}_I = \{\mathbf{A}_I, \mathbf{B}_I\}$ in each layer; Cumulative task statistics $\mathbf{S}^{1:t-1}$ in each layer.
**Output:** The adapted parameters $\mathbf{W}^T$ for each layer in the ViT model.
1 Initialization: $\mathbf{S}^{1:t-1} \leftarrow \mathbf{0}^{D \times D}$.
2 **for** *task* $t = 1$ *to* $T$ **do**

$\quad$ // Before Training: Task-Driven Subspace Decomposition
3 $\quad$ Run $f(\mathbf{W}^{t-1}; \cdot)$ on $\mathcal{D}^t$ and cache the input features of each layer;
4 $\quad$ **for** *each layer in the ViT model* **do**
5 $\quad\quad$ Obtain the cached inputs $\mathbf{X} \in \mathbb{R}^{N^t N^p \times D}$, compute statistics $\mathbf{S}^t = \mathbf{X}^\top \mathbf{X}$;
6 $\quad\quad$ Derive $\mathbf{U}_G$ and $\mathbf{U}_I$ via optimization objectives in Eq. 4 and Eq. 6 using $\mathbf{S}^{1:t-1}$ and $\mathbf{S}^t$;
7 $\quad\quad$ Initialize $\text{LoRA}_G$: $\mathbf{A}_G \leftarrow \mathbf{U}_G^\top$, and $\mathbf{B}_G \leftarrow \mathbf{0}^{D' \times r}$;
8 $\quad\quad$ **if** $t > 1$ **then**
9 $\quad\quad\quad$ Initialize $\text{LoRA}_I$: $\mathbf{A}_I \leftarrow \text{QR}(\mathbf{U}_I^\top)$, and $\mathbf{B}_I \leftarrow \mathbf{0}^{D' \times r}$;
10 $\quad\quad$ **else**
11 $\quad\quad\quad$ Initialize $\text{LoRA}_I$: $\mathbf{A}_I \leftarrow \mathbf{0}^{r \times D}$, and $\mathbf{B}_I \leftarrow \mathbf{0}^{D' \times r}$, freeze $\mathbf{B}_I$ ($\text{LoRA}_I$ ***Disabled***);
12 $\quad\quad$ Freeze backbone weight $\mathbf{W}$, down-projections $\mathbf{A}_G$ and $\mathbf{A}_I$;
13

$\quad$ // During Training: Optimize Up-Projections with GAO
14 $\quad$ **repeat**
15 $\quad\quad$ Sample a mini-batch $\mathcal{B}$ from $\mathcal{D}^t$, split it into two label-disjoint subsets $\mathcal{B}_1$ and $\mathcal{B}_2$;
16 $\quad\quad$ Update trainable parameters by the two-step GAO algorithm in Eq.11 using $\mathcal{B}_1$ and $\mathcal{B}_2$;
17 $\quad$ **until** *Convergence on* $\mathcal{D}^t$
18

$\quad$ // After Training: LoRA Recalibration and Integration
19 $\quad$ **for** *each layer in the ViT model* **do**
20 $\quad\quad$ Compute closed-form diagonal rescaling matrix $\mathbf{\Lambda}_G$ via Eq.13 using $\mathbf{A}_G$, $\mathbf{S}^{1:t-1}$ and $\mathbf{S}^t$;
21 $\quad\quad$ Update backbone weight $\mathbf{W}^t \leftarrow \mathbf{W}^{t-1} + w_G \mathbf{B}_G \mathbf{\Lambda}_G \mathbf{A}_G + \mathbf{B}_I \mathbf{A}_I$;
22 $\quad\quad$ Update accumulated task statistics $\mathbf{S}^{1:t-1} \leftarrow \mathbf{S}^{1:t-1} + \mathbf{S}^t$;
23

---

## B. Theoretical Analysis

### B.1. Proof of Theorem 3.1

**Theorem B.1.** *(Theorem 3.1) Let $\mathcal{L}(\mathbf{Y})$ be a differentiable loss for $\mathbf{Y}$. Fix $\mathbf{A} \in \mathbb{R}^{r \times D}$ and update only $\mathbf{B}$ by one gradient descent step $\mathbf{B}' = \mathbf{B} - \eta \frac{\partial \mathcal{L}}{\partial \mathbf{B}}$. Then the first-order update $\Delta \mathbf{Y}$ of $\mathbf{Y}$ and the corresponding loss decrease is gated by the projected energy $\|\mathbf{A}\mathbf{X}^\top\|_2^2$:*

$$\Delta \mathbf{Y} \triangleq \mathbf{Y}' - \mathbf{Y} = -\eta \|\mathbf{A}\mathbf{X}^\top\|_2^2 \frac{\partial \mathcal{L}}{\partial \mathbf{Y}},$$

$$\mathcal{L}(\mathbf{Y}') - \mathcal{L}(\mathbf{Y}) = -\eta \|\mathbf{A}\mathbf{X}^\top\|_2^2 \left\|\frac{\partial \mathcal{L}}{\partial \mathbf{Y}}\right\|_2^2 + o(\eta).$$

$$(B.1.1)$$

*Moreover, if the rows of $\mathbf{A}$ are orthonormal (i.e., $\mathbf{A}\mathbf{A}^\top = \mathbf{I}_r$), then $\|\mathbf{A}\mathbf{X}^\top\|_2^2$ is exactly the squared Euclidean norm of the projection of $\mathbf{X}$ onto the LoRA input subspace $\text{row}(\mathbf{A})$.*

*Proof.* Denote $\mathbf{g} \triangleq \frac{\partial \mathcal{L}}{\partial \mathbf{Y}} \in \mathbb{R}^{1 \times D'}$, is the gradient of the loss $\mathcal{L}$ with respect to the LoRA output $\mathbf{Y}$, where $\mathbf{Y} = \mathbf{X}(\mathbf{W}^{t-1} + \mathbf{B}\mathbf{A})^\top$. Given that the up-projection $\mathbf{B}$ is updated and the down-projection $\mathbf{A}$ is frozen, the differential $d\mathbf{Y}$ of $\mathbf{Y}$ only depends on $d\mathbf{B}$:

$$d\mathbf{Y} = \mathbf{X}(d\mathbf{B}\,\mathbf{A})^\top = \mathbf{X}\mathbf{A}^\top d\mathbf{B}^\top. \tag{B.1.2}$$

Given the standard first-order differential identity for $\mathcal{L}$, where $d\mathcal{L} = \langle \mathbf{g}, d\mathbf{Y} \rangle = \text{tr}(\mathbf{g}\, d\mathbf{Y}^\top)$ and $\langle \cdot, \cdot \rangle$ denotes the Frobenius inner product. By substituting $d\mathbf{Y}$ with Eq.B.1.2, we obtain the relationship between $d\mathcal{L}$ and $d\mathbf{B}$:

$$d\mathcal{L} = \text{tr}\left(\mathbf{g}\, d\mathbf{B}\, \mathbf{A}\, \mathbf{X}^\top\right) = \text{tr}\left((\mathbf{g}^\top(\mathbf{X}\mathbf{A}^\top))^\top d\mathbf{B}\right). \tag{B.1.3}$$

Meanwhile, the differential of $\mathcal{L}$ with respect to $\mathbf{B}$ satisfies $d\mathcal{L} = \text{tr}\left(\left(\frac{\partial \mathcal{L}}{\partial \mathbf{B}}\right)^\top d\mathbf{B}\right)$, where $\frac{\partial \mathcal{L}}{\partial \mathbf{B}} \in \mathbb{R}^{D' \times r}$. By comparing this formulation with Eq.B.1.3, we can derive the relationship between $\frac{\partial \mathcal{L}}{\partial \mathbf{B}}$ and $\mathbf{g}$, which is formulated as:

$$\frac{\partial \mathcal{L}}{\partial \mathbf{B}} = \mathbf{g}^\top(\mathbf{X}\mathbf{A}^\top) \in \mathbb{R}^{D' \times r}. \tag{B.1.4}$$

It explicitly shows that $\frac{\partial \mathcal{L}}{\partial \mathbf{B}}$ is determined by the loss gradient w.r.t. $\mathbf{Y}$ and the component of $\mathbf{X}$ captured by the subspace induced by down-projection $\mathbf{A}$. Now apply one gradient descent step on $\mathbf{B}$, where $\mathbf{B}' = \mathbf{B} - \eta \frac{\partial \mathcal{L}}{\partial \mathbf{B}}$. Define $\Delta\mathbf{B} \triangleq \mathbf{B}' - \mathbf{B}$ as the the parameter change of $\mathbf{B}$, the induced output change $\Delta\mathbf{Y}$ can be formulated as follows:

$$\Delta\mathbf{Y} = \mathbf{X}(\Delta\mathbf{B}\,\mathbf{A})^\top = -\eta\,\mathbf{X}\left(\frac{\partial \mathcal{L}}{\partial \mathbf{B}}\mathbf{A}\right)^\top = -\eta\,\mathbf{X}\left(\mathbf{g}^\top(\mathbf{X}\mathbf{A}^\top)\mathbf{A}\right)^\top = -\eta\,\mathbf{X}\mathbf{A}^\top\mathbf{A}\mathbf{X}^\top\mathbf{g}. \tag{B.1.5}$$

Since $\mathbf{X}$ is a row input vector and $\mathbf{X}\mathbf{A}^\top\mathbf{A}\mathbf{X}^\top \in \mathbb{R}^{1 \times 1}$ is a scalar, we have $\mathbf{X}\mathbf{A}^\top\mathbf{A}\mathbf{X}^\top = \mathbf{X}\mathbf{A}^\top(\mathbf{A}\mathbf{X}^\top) = \|\mathbf{A}\mathbf{X}^\top\|_2^2$. Then the above expression of $\Delta\mathbf{Y}$ can be rewritten as follows:

$$\Delta\mathbf{Y} = -\eta\,\|\mathbf{A}\mathbf{X}^\top\|_2^2\mathbf{g} = -\eta\,\|\mathbf{A}\mathbf{X}^\top\|_2^2\frac{\partial \mathcal{L}}{\partial \mathbf{Y}}. \tag{B.1.6}$$

Finally, a first-order Taylor expansion of the loss $\mathcal{L}$ around $\mathbf{Y}$ can be formulated as follows:

$$\begin{aligned}
\mathcal{L}(\mathbf{Y}') - \mathcal{L}(\mathbf{Y}) &= \left\langle \frac{\partial \mathcal{L}}{\partial \mathbf{Y}}, \Delta\mathbf{Y} \right\rangle + o(\|\Delta\mathbf{Y}\|) \\
&= -\eta\,\|\mathbf{A}\mathbf{X}^\top\|_2^2\left\|\frac{\partial \mathcal{L}}{\partial \mathbf{Y}}\right\|_2^2 + o(\|\Delta\mathbf{Y}\|),
\end{aligned} \tag{B.1.7}$$

where $o(\|\Delta\mathbf{Y}\|)$ collects higher-order terms in the output change. This completes the proof. $\square$

## B.2. A Brief Derivation of the Optimal Solution $\mathbf{U}_G$ in Eq. 4

The optimization problem in Eq. 4 is formulated as follows:

$$\mathbf{U}_G = \arg\max_{\mathbf{U} \in \mathbb{R}^{D \times r}} \text{tr}\left(\mathbf{U}^\top\mathbf{S}^{1:t-1}\mathbf{U} + \mathbf{U}^\top\mathbf{S}^t\mathbf{U}\right), \quad \text{s.t.} \quad \mathbf{U}^\top\mathbf{U} = \mathbf{I}_r. \tag{B.2.1}$$

We first show that each $\mathbf{S}^i$ is symmetric positive semi-definite. By definition, $\mathbf{S}^i = \mathbf{X}^{i\top}\mathbf{X}^i \in \mathbb{R}^{D \times D}$. For any $D$-dimension vector $\mathbf{z} \in \mathbb{R}^D$, we have the following rule:

$$\mathbf{z}^\top\mathbf{S}^i\mathbf{z} = \mathbf{z}^\top\mathbf{X}^{i\top}\mathbf{X}^i\mathbf{z} = \|\mathbf{X}^i\mathbf{z}\|_2^2 \geq 0, \tag{B.2.2}$$

which implies $\mathbf{S}^i \succeq \mathbf{0}$. Moreover, $\mathbf{S}^i$ is symmetric since $\mathbf{S}^{i\top} = \mathbf{X}^{i\top}\mathbf{X}^i = \mathbf{S}^i$. Therefore, $\mathbf{S}^{1:t-1} + \mathbf{S}^t$ is also symmetric positive semi-definite. Let $[\mathbf{U}_S, \mathbf{\Sigma}_S, \mathbf{V}_S] = SVD(\mathbf{S}^{1:t-1}+\mathbf{S}^t)$, where the singular values in $\mathbf{\Sigma}_S$ are sorted in non-increasing order. Since $\mathbf{S}^{1:t-1} + \mathbf{S}^t$ is symmetric positive semi-definite, its SVD coincides with its eigenvalue decomposition, where $\mathbf{V}_S = \mathbf{U}_S$. Then we have:

$$\mathbf{S}^{1:t-1} + \mathbf{S}^t = \mathbf{U}_S\mathbf{\Sigma}_S\mathbf{V}_S^\top = \mathbf{U}_S\mathbf{\Sigma}_S\mathbf{U}_S^\top, \quad \mathbf{\Sigma}_S = \text{diag}(\sigma_1, \ldots, \sigma_D), \quad \sigma_1 \geq \cdots \geq \sigma_D \geq 0. \tag{B.2.3}$$

Substituting Eq. B.2.3 into the objective yields

$$\text{tr}\big(\mathbf{U}^\top(\mathbf{S}^{1:t-1} + \mathbf{S}^t)\mathbf{U}\big) = \text{tr}\big(\mathbf{U}^\top\mathbf{U}_S\boldsymbol{\Sigma}_S\mathbf{U}_S^\top\mathbf{U}\big) = \text{tr}\big((\mathbf{U}_S^\top\mathbf{U})^\top\boldsymbol{\Sigma}_S(\mathbf{U}_S^\top\mathbf{U})\big). \tag{B.2.4}$$

Define $\mathbf{U}' \triangleq \mathbf{U}_S^\top\mathbf{U} \in \mathbb{R}^{D\times r}$. Since $\mathbf{U}^\top\mathbf{U} = \mathbf{I}_r$ and $\mathbf{U}_S\mathbf{U}_S^\top = \mathbf{I}_D$, we have $\mathbf{U}'^\top\mathbf{U}' = \mathbf{U}^\top\mathbf{U}_S\mathbf{U}_S^\top\mathbf{U} = \mathbf{U}^\top\mathbf{U} = \mathbf{I}_r$, $i.e.$, the columns of $\mathbf{U}'$ are orthonormal. Therefore, Eq.B.2.4 can be rewritten as follows:

$$\text{tr}\big(\mathbf{U}^\top(\mathbf{S}^{1:t-1} + \mathbf{S}^t)\mathbf{U}\big) = \text{tr}\big(\mathbf{U}'^\top\boldsymbol{\Sigma}_S\mathbf{U}'\big) = \sum_{j=1}^{D}\sigma_j\|\mathbf{u}'_j\|_2^2, \tag{B.2.5}$$

where $\mathbf{u}'_j \in \mathbb{R}^r$ denotes the $j$-th row of $\mathbf{U}'$.

For each $j$, we have $\|\mathbf{u}'_j\|_2^2 = \|\mathbf{e}_j^\top\mathbf{U}'\|_2^2 \le \|\mathbf{e}_j\|_2^2\|\mathbf{U}'\|_2^2 = \|\mathbf{U}'\|_2^2$, where $\mathbf{e}_j \in \mathbb{R}^D$ denotes the $j$-th standard basis vector. Meanwhile, since $\mathbf{U}'^\top\mathbf{U}' = \mathbf{I}_r$, we have $\|\mathbf{U}'\|_2^2 = \lambda_{\max}\big(\mathbf{U}'^\top\mathbf{U}'\big) = \lambda_{\max}(\mathbf{I}_r) = 1$, where $\lambda_{\max}(\cdot)$ denotes the operation of taking the largest eigenvalue. It implies $0 \le \|\mathbf{u}'_j\|_2^2 \le 1$ for all $j$.

Moreover, we have $\sum_{j=1}^{D}\|\mathbf{u}'_j\|_2^2 = \|\mathbf{U}'\|_F^2 = \text{tr}(\mathbf{U}'^\top\mathbf{U}') = \text{tr}(\mathbf{I}_r) = r$.

Given $0 \le \|\mathbf{u}'_j\|_2^2 \le 1$ and $\sum_{j=1}^{D}\|\mathbf{u}'_j\|_2^2 = r$, the objective $\sum_{j=1}^{D}\sigma_j\|\mathbf{u}'_j\|_2^2$ is maximized by assigning $\|\mathbf{u}'_j\|_2^2 = 1$ to the largest $r$ singular values:

$$\|\mathbf{u}'_j\|_2^2 = \begin{cases} 1, & j = 1,\ldots,r, \\ 0, & j = r+1,\ldots,D. \end{cases} \tag{B.2.6}$$

The corresponding function value is $\sum_{j=1}^{D}\sigma_j\|\mathbf{u}'_j\|_2^2 = \sum_{j=1}^{r}\sigma_j$. One feasible solution achieving the maximum is $\mathbf{U}' = \begin{bmatrix} \mathbf{I}_r \\ \mathbf{0} \end{bmatrix}$ and $\mathbf{U}_G = \mathbf{U}_{S[:,1:r]}$, $i.e.$, $\mathbf{U}_G$ consists of the top-$r$ singular vectors of matrix $\mathbf{S}^{1:t-1} + \mathbf{S}^t$.

## B.3. Proof of Theorem 3.3

**Theorem B.2** (Theorem 3.3, Closed-form solution for Eq.12)**.** *Let* $\mathbf{S}^t = \mathbf{X}^{t\top}\mathbf{X}^t \in \mathbb{R}^{D\times D}$ *and* $\mathbf{S}^{1:t-1} = \sum_{i=1}^{t-1}\mathbf{X}^{i\top}\mathbf{X}^i \in \mathbb{R}^{D\times D}$, *the optimal solution of Eq.12 is given by:*

$$\mathbf{B}_G^{(j)\star} = \gamma^{(j)\star}\mathbf{B}_G^{(j)} = \frac{\lambda\mathbf{A}_G^{(j)}\mathbf{S}^t\mathbf{A}_G^{(j)\top}}{\mathbf{A}_G^{(j)}(\lambda\mathbf{S}^t + \mathbf{S}^{1:t-1})\mathbf{A}_G^{(j)\top}}\mathbf{B}_G^{(j)}. \tag{B.3.1}$$

*Proof.* For the $j$-th rank-1 unit in LoRA$_G$, we have $\mathbf{A}_G^{(j)} \in \mathbb{R}^{1\times D}$ and $\mathbf{B}_G^{(j)}, \mathbf{B}_G^{(j)\star} \in \mathbb{R}^{D'\times 1}$, and $(\mathbf{B}_G^{(j)\star}\mathbf{A}_G^{(j)})^\top = \mathbf{A}_G^{(j)\top}\mathbf{B}_G^{(j)\star\top}$. The objective function (denoted as $\mathcal{J}(\mathbf{B}_G^{(j)\star})$ in Eq.12 is defined as:

$$\mathcal{J}\Big(\mathbf{B}_G^{(j)\star}\Big) = \lambda\Big\|\mathbf{X}^t(\mathbf{B}_G^{(j)\star}\mathbf{A}_G^{(j)})^\top - \mathbf{X}^t(\mathbf{B}_G^{(j)}\mathbf{A}_G^{(j)})^\top\Big\|_F^2 + \sum_{i=1}^{t-1}\Big\|\mathbf{X}^i(\mathbf{B}_G^{(j)\star}\mathbf{A}_G^{(j)})^\top\Big\|_F^2. \tag{B.3.2}$$

The first term of the objective in Eq.12 can be transformed into follows:

$$\begin{aligned} &\Big\|\mathbf{X}^t(\mathbf{B}_G^{(j)\star}\mathbf{A}_G^{(j)})^\top - \mathbf{X}^t(\mathbf{B}_G^{(j)}\mathbf{A}_G^{(j)})^\top\Big\|_F^2 \\ =&\Big\|\mathbf{X}^t\mathbf{A}_G^{(j)\top}\Big(\mathbf{B}_G^{(j)\star} - \mathbf{B}_G^{(j)}\Big)^\top\Big\|_F^2 \\ =&\text{tr}\bigg(\Big(\mathbf{B}_G^{(j)\star} - \mathbf{B}_G^{(j)}\Big)\mathbf{A}_G^{(j)}\mathbf{X}^{t\top}\mathbf{X}^t\mathbf{A}_G^{(j)\top}\Big(\mathbf{B}_G^{(j)\star} - \mathbf{B}_G^{(j)}\Big)^\top\bigg) \\ =&\text{tr}\bigg(\Big(\mathbf{B}_G^{(j)\star} - \mathbf{B}_G^{(j)}\Big)\mathbf{A}_G^{(j)}\mathbf{S}^t\mathbf{A}_G^{(j)\top}\Big(\mathbf{B}_G^{(j)\star} - \mathbf{B}_G^{(j)}\Big)^\top\bigg). \end{aligned} \tag{B.3.3}$$

Similarly, for the second term in Eq.12, we have:

$$
\begin{aligned}
&\sum_{i=1}^{t-1}\left\|\mathbf{X}^i(\mathbf{B}_G^{(j)\star}\mathbf{A}_G^{(j)})^\top - \mathbf{X}^i\mathbf{0}^{D\times D'}\right\|_F^2 \\
&=\sum_{i=1}^{t-1}\operatorname{tr}\left(\mathbf{B}_G^{(j)\star}\mathbf{A}_G^{(j)}\mathbf{X}^{i\top}\mathbf{X}^i\mathbf{A}_G^{(j)\top}\mathbf{B}_G^{(j)\star\top}\right) \\
&=\operatorname{tr}\left(\mathbf{B}_G^{(j)\star}\mathbf{A}_G^{(j)}\mathbf{S}^{1:t-1}\mathbf{A}_G^{(j)\top}\mathbf{B}_G^{(j)\star\top}\right).
\end{aligned}
\tag{B.3.4}
$$

By substituting Eq.B.3.3 and Eq.B.3.4 into the origin objective and expanding the quadratic term in Eq.B.3.3, we transform the objective function as follows:

$$
\begin{aligned}
\mathcal{J}\left(\mathbf{B}_G^{(j)\star}\right) = {}&\lambda\operatorname{tr}\left(\mathbf{B}_G^{(j)\star}\mathbf{A}_G^{(j)}\mathbf{S}^t\mathbf{A}_G^{(j)\top}\mathbf{B}_G^{(j)\star\top}\right) - 2\lambda\operatorname{tr}\left(\mathbf{B}_G^{(j)\star}\mathbf{A}_G^{(j)}\mathbf{S}^t\mathbf{A}_G^{(j)\top}\mathbf{B}_G^{(j)\top}\right) \\
&+ \lambda\operatorname{tr}\left(\mathbf{B}_G^{(j)}\mathbf{A}_G^{(j)}\mathbf{S}^t\mathbf{A}_G^{(j)\top}\mathbf{B}_G^{(j)\top}\right) + \operatorname{tr}\left(\mathbf{B}_G^{(j)\star}\mathbf{A}_G^{(j)}\mathbf{S}^{1:t-1}\mathbf{A}_G^{(j)\top}\mathbf{B}_G^{(j)\star\top}\right).
\end{aligned}
$$

Since $\mathcal{J}(\mathbf{B}_G^{(j)\star})$ is a convex quadratic function of $\mathbf{B}_G^{(j)\star}$, hence the global optimum is attained at the stationary point $\nabla_{\mathbf{B}_G^{(j)\star}}\mathcal{J} = \mathbf{0}$. We then derive $\nabla_{\mathbf{B}_G^{(j)\star}}\mathcal{J}$ as follows:

$$
\nabla_{\mathbf{B}_G^{(j)\star}}\mathcal{J} = 2\mathbf{B}_G^{(j)\star}\,\mathbf{A}_G^{(j)}(\lambda\mathbf{S}^t + \mathbf{S}^{1:t-1})\mathbf{A}_G^{(j)\top} - 2\lambda\,\mathbf{B}_G^{(j)}\,\mathbf{A}_G^{(j)}\mathbf{S}^t\mathbf{A}_G^{(j)\top}.
\tag{B.3.5}
$$

Setting it to $\mathbf{0}$ and directly solving for $\mathbf{B}_G^{(j)\star}$ yields:

$$
\mathbf{B}_G^{(j)\star} = \frac{\lambda\mathbf{A}_G^{(j)}\mathbf{S}^t\mathbf{A}_G^{(j)\top}}{\mathbf{A}_G^{(j)}(\lambda\mathbf{S}^t + \mathbf{S}^{1:t-1})\mathbf{A}_G^{(j)\top}}\mathbf{B}_G^{(j)} \;\triangleq\; \gamma^{(j)\star}\mathbf{B}_G^{(j)}.
\tag{B.3.6}
$$

$$\square$$

**Discussion.** The above closed-form solution reveals that the optimal merged update for each rank-1 unit is a re-scaling of the current unit, *i.e.*, $\mathbf{B}_G^{(j)\star} = \gamma^{(j)\star}\mathbf{B}_G^{(j)}$. The coefficient $\gamma^{(j)\star}$ is determined by the energy of the current-task covariance $\mathbf{S}^t$ and the accumulated past covariance $\mathbf{S}^{1:t-1}$ along the same input-side direction $\mathbf{A}_G^{(j)}$. Intuitively, this yields a principled trade-off: when $\mathbf{A}_G^{(j)}$ aligns more with the current task, the merge preserves more of the current update. When it also induces large responses on past data, the merge shrinks the update to mitigate interference.

### B.4. Theoretical Analysis of Gradient-Aligned Optimization (GAO)

**Equivalence between GAO and the Explicit Gradient Alignment Objective.** Denote $\mathbf{g}_1 \triangleq \nabla_\theta\mathcal{L}(\theta, \mathcal{B}_1)$ and $\mathbf{g}_2 \triangleq \nabla_\theta\mathcal{L}(\theta, \mathcal{B}_2)$. Consider the following objective for a given $\rho > 0$:

$$
L_\rho(\theta) \triangleq \mathcal{L}\left(\theta - \rho\frac{\mathbf{g}_2}{\|\mathbf{g}_2\|_2^2}, \mathcal{B}_1\right) + \mathcal{L}\left(\theta - \rho\frac{\mathbf{g}_1}{\|\mathbf{g}_1\|_2^2}, \mathcal{B}_2\right).
\tag{B.4.1}
$$

If the inner perturbations $\rho\frac{\mathbf{g}_2}{\|\mathbf{g}_2\|_2^2}$ and $\rho\frac{\mathbf{g}_1}{\|\mathbf{g}_1\|_2^2}$ are treated as constants when computing the outer gradients, then one gradient step on $\mathcal{L}_\rho$ decomposes exactly into the two GAO updates:

$$
\theta \leftarrow \theta - \mathrm{lr}\times\nabla_\theta\mathcal{L}\left(\theta - \rho\frac{\mathbf{g}_2}{\|\mathbf{g}_2\|_2^2}, \mathcal{B}_1\right), \qquad \theta \leftarrow \theta - \mathrm{lr}\times\nabla_\theta\mathcal{L}\left(\theta - \rho\frac{\mathbf{g}_1}{\|\mathbf{g}_1\|_2^2}, \mathcal{B}_2\right).
\tag{B.4.2}
$$

Assuming $\mathcal{L}(\cdot, \mathcal{B}_1)$ and $\mathcal{L}(\cdot, \mathcal{B}_2)$ are twice differentiable, a first-order Taylor expansion around $\theta$ gives

$$
\mathcal{L}\left(\theta - \rho\frac{\mathbf{g}_2}{\|\mathbf{g}_2\|_2^2}, \mathcal{B}_1\right) = \mathcal{L}(\theta, \mathcal{B}_1) - \rho\,\mathbf{g}_1^\top\frac{\mathbf{g}_2}{\|\mathbf{g}_2\|_2^2} + \mathcal{O}(\rho^2),
\tag{B.4.3}
$$

$$
\mathcal{L}\left(\theta - \rho\frac{\mathbf{g}_1}{\|\mathbf{g}_1\|_2^2}, \mathcal{B}_2\right) = \mathcal{L}(\theta, \mathcal{B}_2) - \rho\,\mathbf{g}_2^\top\frac{\mathbf{g}_1}{\|\mathbf{g}_1\|_2^2} + \mathcal{O}(\rho^2).
\tag{B.4.4}
$$

Summing (B.4.3)–(B.4.4) yields the first-order equivalence of our objective:

$$\mathcal{L}_\rho(\theta) = \mathcal{L}(\theta, \mathcal{B}) - \rho\,\mathbf{g}_1^\top \mathbf{g}_2\left(\frac{1}{\|\mathbf{g}_1\|_2^2} + \frac{1}{\|\mathbf{g}_2\|_2^2}\right) + \mathcal{O}(\rho^2). \tag{B.4.5}$$

Equation (B.4.5) shows that GAO implements a stochastic first-order optimization of gradient alignment: minimizing $\mathcal{L}_\rho$ directly promotes larger cross-group inner products $\mathbf{g}_i^\top \mathbf{g}_j$.

**Theoretical Connection between GAO and the Anti-Interference Ability.** We then analyze the connection between encouraging large gradient similarities in Eq.B.4.5 and tightening the anti-interference ability of the learned parameters. Let $\mathcal{D}$ be a mini-batch from the current task and let $\{\mathcal{D}_i\}_{i=1}^N$ be a partition of $\mathcal{D}$ into $N$ label-disjoint groups, where $\mathcal{D} = \mathcal{D}_1 \cup \cdots \cup \mathcal{D}_N$. Denote gradients of the $i$-th group $\mathcal{D}_i$ as $\mathbf{g}_i \triangleq \nabla_\theta \mathcal{L}(\theta, \mathcal{D}_i)$, and define the gradient cone as:

$$\mathcal{C} \triangleq \Big\{ \sum_{i=1}^N \alpha_i \mathbf{g}_i : \; \alpha_i \geq 0 \Big\}. \tag{B.4.6}$$

Considering the gradient diversity provided by sufficient within-task samples and the high similarity between incremental classification tasks, we assume that a possible feature gradient perturbation can be decomposed into a conic combination of the current-task group gradients and a bounded residual:

$$\mathbf{g}_{\text{fut}} = \mathbf{c} + \boldsymbol{\xi} = \sum_{i=1}^N \beta_i \mathbf{g}_i + \boldsymbol{\xi}, \qquad \mathbf{c} \in \mathcal{C}, \qquad \|\boldsymbol{\xi}\|_2 \leq \varepsilon. \tag{B.4.7}$$

When learning a future task, consider one update $\theta^+ = \theta - \eta\,\mathbf{g}_{\text{fut}}$ with $\eta > 0$. Assume $\mathcal{L}(\mathcal{D}, \theta)$ satisfies local linearity, then the loss change can be formulated as follows:

$$\mathcal{L}(\mathcal{D}, \theta^+) - \mathcal{L}(\mathcal{D}, \theta) \leq -\eta \nabla_\theta \mathcal{L}(\mathcal{D}, \theta)^\top \mathbf{g}_{\text{fut}} + \mathcal{O}(\eta^2 \|\mathbf{g}_{\text{fut}}\|_2^2). \tag{B.4.8}$$

Since the gradient $\nabla_\theta \mathcal{L}(\mathcal{D}, \theta)$ computed by $\mathcal{D}$ lies in the gradient cone $\mathcal{C}$, there exist coefficients $\{\gamma_i\}_{i=1}^N$ with $\gamma_i \geq 0$ such that $\nabla_\theta \mathcal{L}(\mathcal{D}, \theta) = \sum_{i=1}^N \gamma_i \mathbf{g}_i$. The loss change is bounded by the following formulation:

$$\mathcal{L}(\mathcal{D}, \theta^+) - \mathcal{L}(\mathcal{D}, \theta) \leq -\eta \sum_{i=1}^N \sum_{j=1}^N \gamma_i \beta_j\,\mathbf{g}_i^\top \mathbf{g}_j \; - \; \eta \sum_{i=1}^N \gamma_i\,\mathbf{g}_i^\top \boldsymbol{\xi} + \mathcal{O}(\eta^2 \|\mathbf{g}_{\text{fut}}\|_2^2). \tag{B.4.9}$$

**Discussion.** Theorem B.4 reveals how within-task gradient alignment improves the anti-interference ability. The leading term in (B.4.9) is a weighted sum of pairwise inner products $\mathbf{g}_i^\top \mathbf{g}_j$ between label-disjoint groups. When these cross-group correlations $\mathbf{g}_i^\top \mathbf{g}_j$ are consistently positive and sufficiently large, any future update whose gradient direction admits the cone decomposition (B.4.7) tends to induce a smaller increase or even a decrease on the current-task loss. Therefore, encouraging shared directions across groups effectively enlarges the safe set of update directions in which the current task is less likely to be harmed, improving robustness to future optimization steps.

**Experimental Validation.** To demonstrate the effectiveness of our GAO algorithm, we visualize the training loss curves of the first task $\mathcal{D}^1$ and the new task $\mathcal{D}^t$ across incremental sessions, as shown in Fig.7. As the task increases, the training loss on the first task $\mathcal{D}^1$ steadily increases due to interference from subsequent updates. GAO consistently mitigates this drift on both datasets, supporting stronger robustness against interference from updates of future tasks. Meanwhile, GAO also yields a lower loss on the new task $\mathcal{D}^t$, indicating more effective within-task optimization. These trends agree with Theorem B.4: by enlarging the safe set of update directions under the cone model in (B.4.7), GAO improves robustness of learned directions and alleviates the forgetting of previous knowledge.

## C. More Detailed Experimental Results

### C.1. Hyper-Parameter Analysis

We conduct experiments to investigate how several key hyper-parameters affect performance on 10S-ImageNetR and 10S-ImageNetA, $i.e.$, $w_G$ in Eq.1, $\lambda$ in Eq.12, and $\rho_{\max}$ for GAO in Eq.11. The experimental results are shown in Fig.8.

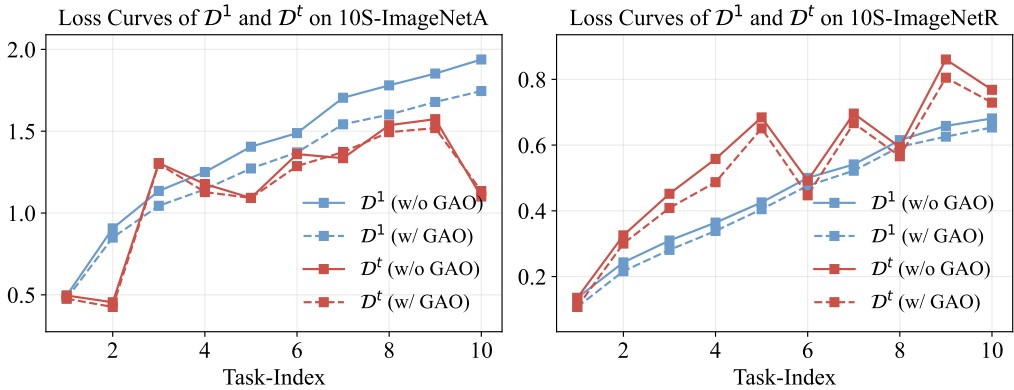

*Figure 7.* Training loss curves of the first task $\mathcal{D}^1$ and the new task $\mathcal{D}^t$ across incremental sessions on 10S-ImageNetR and 10S-ImageNetA.

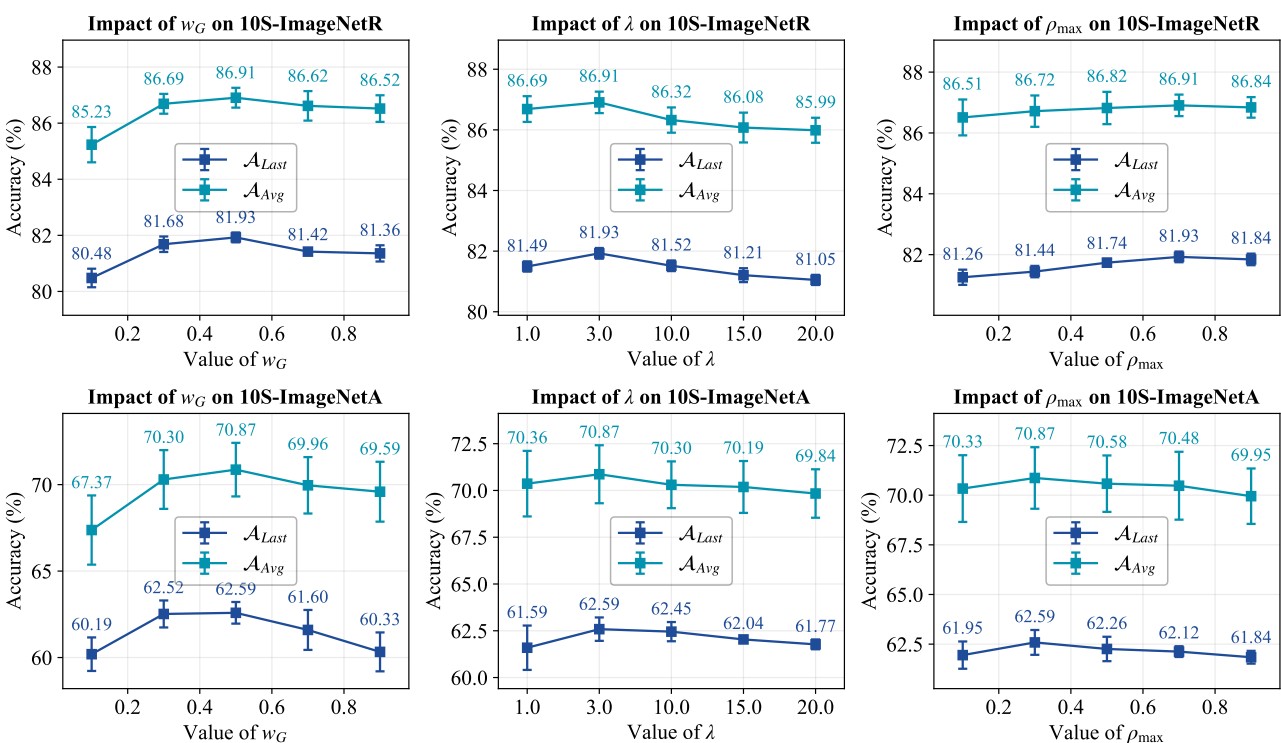

*Figure 8.* Impact of the hyper-parameters $w_G$, which balances the contributions of the two LoRA branches in Eq. 3, $\lambda$, which trades off performance between past and new tasks in Eq. 12, and $\rho_{\max}$, which controls the inner gradient magnitude in GAO in Eq. 11, on 10S-ImageNetR and 10S-ImageNetA. All results are averaged over three random seeds, and we report the mean with error bars.

**Impact of $w_G$.** The hyper-parameter $w_G$ controls the relative contribution of the general branch LoRA$_G$ and the isolated branch LoRA$_I$ in forward propagation, thus directly adjusting the balance between cross-task knowledge sharing and task-specific updates. As shown in Fig.8, a too-small $w_G$ (*e.g.*, 0.1) is sub-optimal since the model is dominated by LoRA$_I$ and cannot sufficiently exploit task-shared directions, highlighting the necessity of learning general knowledge. When $w_G \geq 0.7$, the performance decreases notably because the update becomes overly dominated by the general branch, which fails to learn enough novel directions during continual adaptation while inducing inevitable feature drifts of old tasks. This trend is consistent with Fig.6, where $\mathbf{A}_G$ already captures most of the input component, thus it is necessary to down-weight LoRA$_G$ with a proper coefficient $0.0 \leq w_G \leq 1.0$. According to the results, we set $w_G = 0.5$.

**Impact of $\lambda$.** The hyper-parameter $\lambda$ in controls the trade-off in rescaling the general branch between preserving the current-task performances and limiting feature drift on previous tasks. As shown in Fig. 8, although $\lambda = 1.0$ assigns equal weight to the new task and the accumulated past tasks, it overly shrinks the general update and insufficiently adapts each new task, which accumulates larger approximation errors across sessions and results in sub-optimal performances. In contrast, a relatively large $\lambda$ (*e.g.*, $\lambda = 10.0$) does not cause a notable degradation, suggesting that updates along the general subspace are inherently transferable, so aggressive adaptation on later tasks does not substantially harm performances on earlier tasks. Overall, the results are insensitive within $3.0 \leq \lambda \leq 15.0$. Based on the results, we set $\lambda = 3.0$ in our method.

**Impact of $\rho_{\max}$.** The hyper-parameter $\rho_{\max}$ bounds the randomized inner-step scale $\rho \sim \text{U}(0, \rho_{\max})$ in Eq. 11, controlling the perturbation magnitude used to couple gradients in GAO. When $\rho_{\max}$ is small, the perturbation is negligible, so GAO reduces to standard training on the two label-disjoint subsets, minimizing the sum of their cross-entropy losses. In contrast, an overly large $\rho_{\max}$ can push the perturbed parameter outside the local linear region where the implicit alignment approximation is reliable, introducing higher-order effects and harming optimization stability. Based on the results, we set $\rho_{\max} = 0.7$ for ImageNetR and $\rho_{\max} = 0.3$ for the other datasets.

### C.2. Effect of Different LoRA Ranks on LoDA Performance

*Table 7.* Analysis of the effect of rank on additional parameters, storage and computational overhead, and performance on 20S-ImageNetR. We compare our LoDA with different ranks against representative CL methods.

| Methods | Venue | Training Stage (Per Sample) | | | Inference Stage (Per Sample) | | 20s-ImageNetR | |
| --- | --- | --- | --- | --- | --- | --- | --- | --- |
| | | Extra Memory | Trainable Params | Training GFLOPs | Extra Params | Inference GFLOPs | $\mathcal{A}_{Last} \uparrow$ | $\mathcal{A}_{Avg}$ |
| **Methods based on the ViT-21K model (86M).** | | | | | | | | |
| CODA (Smith et al., 2023) | CVPR-23 | 15.4MB | 3.84M (4.46%) | 35.17 | 3.84M (4.46%) | 35.17 | $69.96_{\pm 0.50}$ | $75.34_{\pm 0.85}$ |
| SLCA (Zhang et al., 2023) | ICCV-23 | 450.6MB | 86M (100%) | 17.60 | 0.00M (0.00%) | 17.60 | $74.63_{\pm 1.55}$ | $79.92_{\pm 1.29}$ |
| SSIAT (Tan et al., 2024) | CVPR-24 | 450.6MB | 1.19M (1.38%) | 17.81 | 1.19M (1.38%) | 17.81 | $75.67_{\pm 0.24}$ | $82.30_{\pm 0.36}$ |
| InfLoRA (Liang & Li, 2024) | CVPR-24 | 11.3MB | 0.37M (0.43%) | 17.67 | 0.00M (0.00%) | 17.67 | $71.01_{\pm 0.45}$ | $77.28_{\pm 0.45}$ |
| VPT-NSP (Lu et al., 2024) | NIPS-24 | 28.4MB | 0.09M (0.10%) | 17.72 | 0.09M (0.10%) | 17.72 | $75.69_{\pm 0.61}$ | $81.87_{\pm 0.59}$ |
| SD-LoRA (Wu et al., 2025b) | ICLR-25 | 14.1MB | 0.37M (0.43%) | 20.51 | 0.00M (0.00%) | 17.72 | $75.26_{\pm 0.37}$ | $80.22_{\pm 0.72}$ |
| VPT-CPG (Lu et al., 2025) | AAAI-25 | 28.4MB | 1.66M (1.93%) | 17.72 | 1.66M (1.93%) | 17.72 | $75.33_{\pm 0.37}$ | $80.22_{\pm 0.72}$ |
| **LoDA–r=4 (Ours)** | – | 27.0MB | 0.22M (0.25%) | 17.67 | 0.00M (0.00%) | 17.60 | $\mathbf{78.13_{\pm 0.33}}$ | $\mathbf{84.56_{\pm 0.53}}$ |
| **LoDA–r=8 (Ours)** | – | 27.0MB | 0.44M (0.51%) | 17.71 | 0.00M (0.00%) | 17.60 | $\mathbf{78.47_{\pm 0.61}}$ | $\mathbf{84.76_{\pm 0.35}}$ |
| **LoDA–r=16 (Ours)** | – | 27.0MB | 0.88M (1.02%) | 17.83 | 0.00M (0.00%) | 17.60 | $\mathbf{79.11_{\pm 0.39}}$ | $\mathbf{84.85_{\pm 0.45}}$ |
| **LoDA–r=32 (Ours)** | – | 27.0MB | 1.77M (2.06%) | 18.06 | 0.00M (0.00%) | 17.60 | $\mathbf{78.96_{\pm 0.23}}$ | $\mathbf{84.94_{\pm 0.17}}$ |
| **LoDA–r=64 (Ours)** | – | 27.0MB | 3.53M (4.10%) | 18.52 | 0.00M (0.00%) | 17.60 | $\mathbf{78.93_{\pm 0.05}}$ | $\mathbf{84.91_{\pm 0.59}}$ |

We evaluate the impact of rank in LoDA on the number of additional parameters, storage and computational overhead, and performance on 20S-ImageNetR. We also compare LoDA with different ranks against representative CL baselines under the ViT-21K backbone (Dosovitskiy et al., 2021). Specifically, we vary the LoRA rank $r \in \{4, 8, 16, 32, 64\}$, and denote the corresponding variant as "LoDA-r=$r$". The results are reported in Tab. 7. Overall, LoDA consistently outperforms existing PEFT-based CL methods across all ranks, while introducing only a small number of trainable parameters and modest training compute overhead. Notably, LoDA incurs ***zero extra parameters and overhead at inference*** by merging both branches into the backbone, keeping inference GFLOPs fixed at the backbone cost (17.60). Moreover, LoDA ***avoids storing past-task models for model expansion, regularization, or knowledge distillation.*** Its storage cost comes from a single task statistics matrix of size $D \times D$ in each ViT layer (with $D = 768$ for ViT-21K), making the memory footprint lightweight and scalable as the number of tasks grows.

As shown in Tab. 7, increasing $r$ from 4 to 16 steadily improves accuracy (e.g., $\mathcal{A}_{last}$ rises from 78.13 to 79.11), suggesting that a moderate rank provides sufficient capacity to capture both transferable and task-adaptive directions. Beyond this point, the performance gains saturate, where larger ranks (e.g., $r = 64$) achieve comparable results but incur noticeably higher training compute (training GFLOPs increase from 17.83 at $r = 16$ to 18.52 at $r = 64$) and more trainable parameters. This indicates that LoDA is not overly sensitive to rank and can reach strong performance under small ranks. We set $r = 32$ in all other experiments as a practical trade-off that maintains near-saturated accuracy while keeping the overhead modest.

*Table 8.* Experimental results on ImageNetR, CIFAR100 and UCF datasets using CLIP as the backbone. All methods are initialized with the CLIP pre-trained on LAION-400M *without experience replay* for fair comparisons. PROOF† represents our reproduced results of PROOF (Zhou et al., 2025) without experience replay. The best results are in red and the second highest results are in blue.

| Method | ImageNetR | | | | CIFAR100 | | | | UCF | | | |
|---|---|---|---|---|---|---|---|---|---|---|---|---|
| | B0 Inc10 | | B50 Inc10 | | B0 Inc10 | | B50 Inc10 | | B0 Inc20 | | B100 Inc20 | |
| | $\mathcal{A}_{Avg}$ | $\mathcal{A}_{Last}$ | $\mathcal{A}_{Avg}$ | $\mathcal{A}_{Last}$ | $\mathcal{A}_{Avg}$ | $\mathcal{A}_{Last}$ | $\mathcal{A}_{Avg}$ | $\mathcal{A}_{Last}$ | $\mathcal{A}_{Avg}$ | $\mathcal{A}_{Last}$ | $\mathcal{A}_{Avg}$ | $\mathcal{A}_{Last}$ |
| CoOp (Zhou et al., 2022b) | 60.73 | 37.52 | 54.20 | 39.77 | 47.00 | 24.24 | 41.23 | 24.12 | 47.85 | 33.46 | 42.02 | 24.74 |
| ZS-CLIP (Radford et al., 2021) | 83.37 | 77.17 | 79.57 | 77.17 | 81.81 | 71.38 | 76.49 | 71.38 | 75.50 | 67.64 | 71.44 | 67.64 |
| L2P (Wang et al., 2022b) | 75.97 | 66.52 | 72.82 | 66.77 | 82.74 | 73.03 | 81.14 | 73.61 | 86.34 | 76.43 | 83.95 | 76.62 |
| DualPrompt (Wang et al., 2022a) | 76.21 | 66.65 | 73.22 | 67.58 | 81.63 | 72.44 | 80.12 | 72.57 | 85.21 | 75.82 | 84.31 | 76.35 |
| CODA-Prompt (Smith et al., 2023) | 77.69 | 68.95 | 73.71 | 68.05 | 82.43 | 73.43 | 78.69 | 71.58 | 87.76 | 80.14 | 83.04 | 75.03 |
| SimpleCIL (Zhou et al., 2024) | 81.06 | 74.48 | 76.84 | 74.48 | 84.15 | 76.63 | 80.20 | 76.63 | 90.44 | 85.68 | 88.12 | 85.68 |
| RAPF (Huang et al., 2024) | 86.28 | 79.62 | 84.03 | 79.21 | 86.19 | 79.04 | 82.35 | 78.25 | 92.28 | 80.33 | 90.31 | 81.55 |
| PROOF† (Zhou et al., 2025) | 82.69 | 77.25 | 80.56 | 77.03 | 84.88 | 76.29 | 81.85 | 75.93 | 87.04 | 81.47 | 86.15 | 81.85 |
| **LoDA (Ours)** | 87.32 | 82.65 | 85.53 | 83.00 | 88.86 | 81.72 | 86.77 | 83.82 | 94.55 | 90.75 | 94.87 | 92.00 |

*Table 9.* Experimental results on Aircraft, Cars and CUB datasets using CLIP as the backbone. All methods are initialized with the CLIP pre-trained on LAION-400M *without experience replay* for fair comparisons. PROOF† represents our reproduced results of PROOF (Zhou et al., 2025) without experience replay. The best results are in red and the second highest results are in blue.

| Method | Aircraft | | | | Cars | | | | CUB | | | |
|---|---|---|---|---|---|---|---|---|---|---|---|---|
| | B0 Inc10 | | B50 Inc10 | | B0 Inc10 | | B50 Inc10 | | B0 Inc20 | | B100 Inc20 | |
| | $\mathcal{A}_{Avg}$ | $\mathcal{A}_{Last}$ | $\mathcal{A}_{Avg}$ | $\mathcal{A}_{Last}$ | $\mathcal{A}_{Avg}$ | $\mathcal{A}_{Last}$ | $\mathcal{A}_{Avg}$ | $\mathcal{A}_{Last}$ | $\mathcal{A}_{Avg}$ | $\mathcal{A}_{Last}$ | $\mathcal{A}_{Avg}$ | $\mathcal{A}_{Last}$ |
| CoOp (Zhou et al., 2022b) | 14.54 | 7.14 | 13.05 | 7.77 | 36.46 | 21.65 | 37.40 | 20.87 | 27.61 | 8.57 | 24.03 | 10.14 |
| ZS-CLIP (Radford et al., 2021) | 26.66 | 17.22 | 21.70 | 17.22 | 82.60 | 76.37 | 78.32 | 76.37 | 74.38 | 63.06 | 67.96 | 63.06 |
| L2P (Wang et al., 2022b) | 47.19 | 28.29 | 44.07 | 32.13 | 76.63 | 61.82 | 76.37 | 65.64 | 70.87 | 57.93 | 75.64 | 66.12 |
| DualPrompt (Wang et al., 2022a) | 44.30 | 25.83 | 46.07 | 33.57 | 76.26 | 62.94 | 76.88 | 67.55 | 69.89 | 57.46 | 74.40 | 64.84 |
| CODA-Prompt (Smith et al., 2023) | 45.98 | 27.69 | 45.14 | 32.28 | 80.21 | 66.47 | 75.06 | 64.19 | 73.12 | 62.98 | 73.95 | 62.21 |
| SimpleCIL (Zhou et al., 2024) | 59.24 | 48.09 | 53.05 | 48.09 | 92.04 | 86.85 | 88.96 | 86.85 | 83.81 | 77.52 | 79.75 | 77.52 |
| RAPF (Huang et al., 2024) | 50.38 | 23.61 | 40.47 | 25.44 | 82.89 | 62.85 | 75.87 | 63.19 | 79.09 | 62.77 | 72.82 | 62.93 |
| PROOF† (Zhou et al., 2025) | 47.33 | 39.03 | 54.35 | 45.36 | 86.87 | 84.06 | 88.68 | 82.66 | 79.94 | 72.60 | 76.43 | 71.80 |
| **LoDA (Ours)** | 71.53 | 60.97 | 67.63 | 61.81 | 94.27 | 90.57 | 91.80 | 90.40 | 86.84 | 80.20 | 82.83 | 79.26 |

## C.3. Experimental Results Using the CLIP Model as the Backbone

To demonstrate that our LoDA is not tied to a specific architecture and can generalize to other pre-trained models, we further evaluate it using the representative vision-language model CLIP. The results are shown in Tab.8 and Tab.9.

**Datasets and Evaluation Metrics.** Following prior CLIP-based continual learning methods (Zhou et al., 2025; Huang et al., 2024), we evaluate our method on six benchmarks, namely ImageNetR (Hendrycks et al., 2021a), CIFAR100 (Krizhevsky et al., 2009), UCF (Soomro et al., 2012), Aircraft (Maji et al., 2013), Cars (Krause et al., 2013), and CUB (Wah et al., 2011). ImageNet-R consists of 30,000 images from 200 ImageNet categories that contains artistic renditions (*e.g.*, paintings, cartoons, and sketches). CIFAR100 is a natural image classification benchmark with 60,000 $32 \times 32$ images from 100 object categories. UCF101 is an action recognition dataset consisting of videos from 101 human action classes. Aircraft, Cars and CUB are typical fine-grained classification benchmarks. Aircraft consists of 10,200 high-resolution aircraft photographs from 102 model-variant classes. Cars contains 16,185 car images from 196 classes. CUB is a bird dataset with 11,788 images from 200 bird species. Specifically, we sample (a subset of) 100 classes from Aircraft, Cars and UCF to ease data split. The dataset splits are denoted as "B-x Inc-y", where x represents the number of classes in the first stage, and y represents the number of new classes in each subsequent task. x = 0 means each task contains y classes.

**Implementation Details of LoDA based on the CLIP model.** We implement the proposed LoDA on top of a simple yet effective baseline, *i.e.*, SimpleCIL (Zhou et al., 2024). SimpleCIL keeps the pre-trained model frozen and performs continual adaptation by constructing a prototypical classifier, whose class weights are set to the mean features computed from downstream data in the frozen embedding space, without any additional optimization. Building upon SimpleCIL, we maintain a frozen pre-trained CLIP $f_{\mathrm{pt}}(\cdot)$ with its prototypical classifier $\mathbf{W}_{\mathrm{pt}}$, while sequentially tuning another CLIP $f_{\mathrm{ft}}(\cdot)$

using our LoDA framework. The fine-tuned network is trained with the standard CLIP-style image–text matching loss (Zhou et al., 2025) using the textual classifier $\mathbf{W}_{\text{text}}$. During inference, given an input $\mathbf{x}$, we combine the scores from the frozen and fine-tuned models to produce the final prediction:

$$\mathbf{p} = \alpha \mathbf{W}_{\text{pt}}^{\top} f_{\text{pt}}(\mathbf{x}) + (1 - \alpha) \mathbf{W}_{\text{text}}^{\top} f_{\text{ft}}(\mathbf{x}). \tag{C.3.1}$$

**Comparison methods.** We evaluate LoDA against a broad set of representative CLIP-based baselines. These include prompt-learning methods, namely CoOp (Zhou et al., 2022b), L2P (Wang et al., 2022b), DualPrompt (Wang et al., 2022a), and CODA-Prompt (Smith et al., 2023); CLIP-tailored continual learning approaches such as RAPF (Huang et al., 2024) and PROOF (Zhou et al., 2025); a training-free baseline SimpleCIL (Zhou et al., 2024); and the zero-shot CLIP model (Radford et al., 2021), denoted as ZS-CLIP. For a fair comparison, we follow PROOF (Zhou et al., 2025) and use the CLIP backbone pre-trained on LAION-400M across all experiments.

**Results and Analysis.** As shown in Tab.8 and Tab.9, our LoDA achieves best performances across these datasets, demonstrating its strong ability to generalize to different pre-trained models and different downstream tasks. Notably, several fine-tuning based baselines underperform even the zero-shot CLIP and the training-free SimpleCIL, indicating that continual fine-tuning readily degrades the pre-trained representations. In contrast, LoDA preserves the strength of the pre-trained CLIP while enabling effective adaptation by constraining updates to decomposed general and isolated subspaces, resulting in superior performances. Moreover, compared to existing CLIP-based methods, LoDA has two main advantages: (1) Unlike Gaussian sampling-based feature replay methods such as RAPF (Huang et al., 2024) that maintain per-class distribution statistics, LoDA is replay-free and avoids any class-wise memory that scales linearly with the growing label space, while is more efficient; (2) Instead of only adding a tunable head after CLIP while freezing its core weights, LoDA achieves layer-wise continual fine-tuning, thereby injecting new knowledge into the backbone itself.

## C.4. Experimental Results Using Self-Supervised Pre-Trained Backbones

*Table 10.* Results of existing methods under various pre-trained models. Here, DINO-1k and iBOT-1k denote that the frozen backbone is pre-trained on ImageNet-1k through DINO and iBOT pre-training algorithms, respectively. The best results are in red and the second highest results are in blue.

| PTM | Method | Venue | 10S-ImageNetR $\mathcal{A}_{Last}$ | 10S-ImageNetR $\mathcal{A}_{Avg}$ |
|---|---|---|---|---|
| DINO-1k | L2P (Wang et al., 2022b) | CVPR'22 | $56.71_{\pm0.12}$ | $63.59_{\pm0.21}$ |
| | DualPrompt (Wang et al., 2022a) | ECCV'22 | $60.23_{\pm0.42}$ | $66.57_{\pm0.25}$ |
| | CODAPrompt (Smith et al., 2023) | CVPR'23 | $64.02_{\pm0.68}$ | $71.50_{\pm0.42}$ |
| | LAE (Gao et al., 2023) | ICCV'23 | $61.03_{\pm0.27}$ | $69.89_{\pm0.15}$ |
| | HiDe-Prompt (Wang et al., 2024) | NIPS'23 | $68.11_{\pm0.18}$ | $71.70_{\pm0.01}$ |
| | InfLoRA (Liang & Li, 2024) | CVPR'24 | $68.31_{\pm0.28}$ | $76.15_{\pm0.05}$ |
| | VPT-NSP (Lu et al., 2024) | NIPS'24 | $68.96_{\pm0.94}$ | $76.22_{\pm0.56}$ |
| | SD-LoRA (Wu et al., 2025b) | ICLR'25 | $69.78_{\pm0.63}$ | $75.45_{\pm0.35}$ |
| | CoSO (Cheng et al., 2025) | NIPS'25 | $71.60_{\pm0.44}$ | $79.28_{\pm0.16}$ |
| | Seq-LoRA (Baseline) | – | $54.87_{\pm0.47}$ | $69.27_{\pm0.17}$ |
| | **LoDA (Ours)** | – | $73.66_{\pm0.21}$ | $80.51_{\pm0.68}$ |
| iBOT-1k | L2P (Wang et al., 2022b) | CVPR'22 | $60.80_{\pm0.35}$ | $66.58_{\pm0.28}$ |
| | DualPrompt (Wang et al., 2022a) | ECCV'22 | $63.78_{\pm0.38}$ | $68.88_{\pm0.16}$ |
| | CODAPrompt (Smith et al., 2023) | CVPR'23 | $68.02_{\pm0.48}$ | $74.28_{\pm0.47}$ |
| | LAE (Gao et al., 2023) | ICCV'23 | $64.14_{\pm0.29}$ | $72.59_{\pm0.22}$ |
| | HiDe-Prompt (Wang et al., 2024) | NIPS'23 | $71.33_{\pm0.21}$ | $73.62_{\pm0.13}$ |
| | InfLoRA (Liang & Li, 2024) | CVPR'24 | $71.84_{\pm0.09}$ | $78.29_{\pm0.09}$ |
| | VPT-NSP (Lu et al., 2024) | NIPS'24 | $73.25_{\pm0.78}$ | $79.65_{\pm0.63}$ |
| | Seq-LoRA (Baseline) | – | $56.28_{\pm0.52}$ | $71.15_{\pm0.85}$ |
| | **LoDA (Ours)** | – | $76.83_{\pm0.27}$ | $82.83_{\pm0.57}$ |

To further validate the generality of our framework on other backbones, we replace the supervised ViT-B/16 (Dosovitskiy

et al., 2021) backbone pre-trained on ImageNet-21K (Russakovsky et al., 2015) with self-supervised models pre-trained on ImageNet-1K using DINO (Caron et al., 2021) and iBOT (Zhou et al., 2022a), respectively. The performances are reported in Tab. 10 and the task-by-task accuracy curves are shown in Fig. 9. As shown in Tab. 10, LoDA consistently achieves the best performance compared to existing methods, outperforming the state-of-the-arts CoSO by +2.06% $\mathcal{A}_{Last}$ under DINO-1K and VPT-NSP by +3.58% $\mathcal{A}_{Last}$ under iBOT-1K. Meanwhile, it brings large improvements over the sequential tuning baseline "Seq-LoRA", with +18.79% and +20.55% gains on $\mathcal{A}_{Last}$ under DINO-1k and iBOT-1k, respectively. Fig. 9 further supports this advantage. Compared with Seq-LoRA, LoDA exhibits a substantially slower accuracy decay as tasks increase, demonstrating its effectiveness on mitigating catastrophic forgetting. Although the overall CL performance with self-supervised pre-training is lower than that with the supervised backbone, LoDA still achieve superior performances across both DINO and iBOT, highlighting its generality and robustness to different backbones.

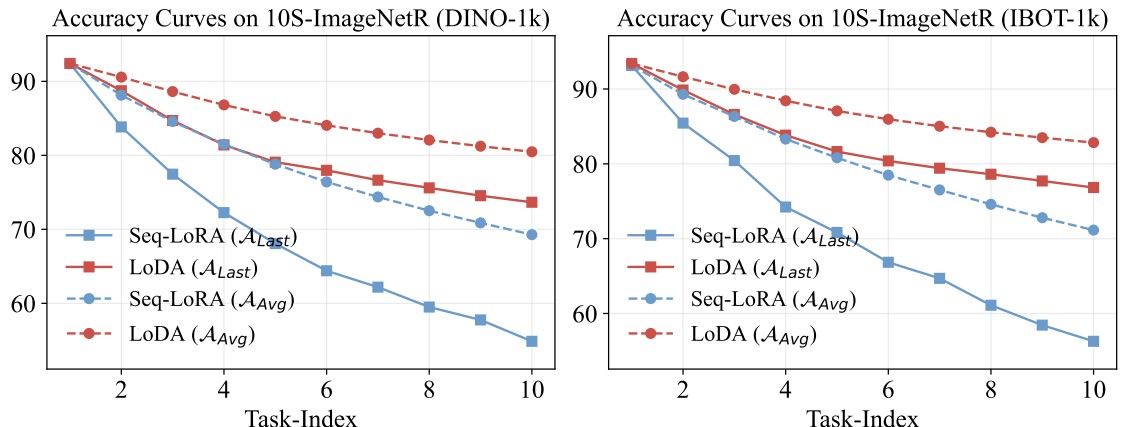

*Figure 9.* Task-by-task accuracy curve on 10S-ImageNetR using different self-supervised pre-trained models.

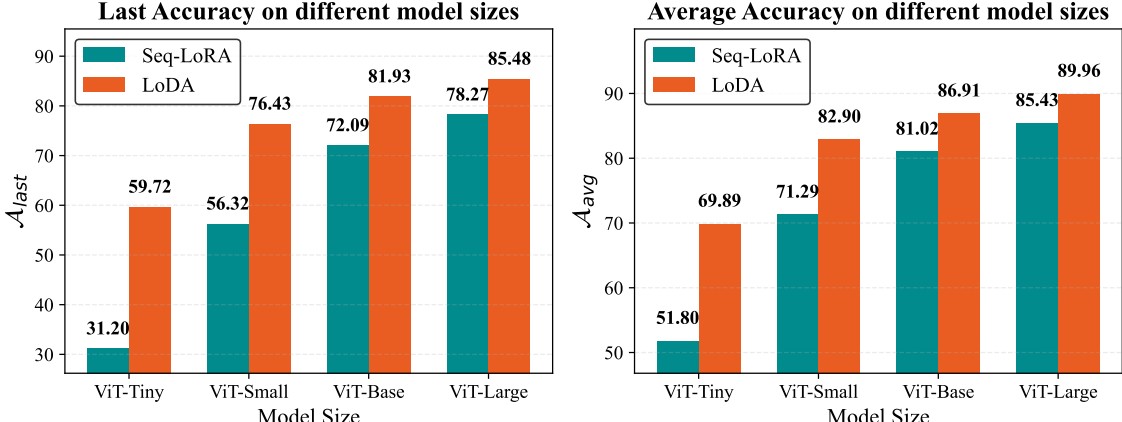

*Figure 10.* Performances of sequential tuning and LoDA using ViT models with different sizes on 10S-ImageNetR.

### C.5. Experimental Results Using Vision Transformers with Different Sizes

To further evaluate the generality and scalability of our LoDA, we conduct experiments using ViT backbones with different sizes, including ViT-Tiny ($D = 192$), ViT-Small ($D = 384$), ViT-Base ($D = 768$), and ViT-Large ($D = 1024$) on the 10S-ImageNetR benchmark. We evaluate the sequential tuning baseline (denoted as "Seq-LoRA") and our method, the results are shown in Fig.10. As shown in Fig.10, our LoDA delivers consistent and clear improvements over Seq-LoRA across 4 model sizes on both $\mathcal{A}_{Last}$ and $\mathcal{A}_{Avg}$, with gains ranging from +7.21% to +28.52% on $\mathcal{A}_{Last}$. Notably, the advantages are more significant on smaller models (*e.g.*, ViT-Tiny). It is because their weak pre-trained representations require larger magnitudes of parameter updates, which better highlights the effectiveness of improving the stability–plasticity trade-off by constraining updates within structured subspaces. Overall, our LoDA is a scalable and model size-agnostic CL framework that brings consistent performance improvements.

## C.6. Visualization of the Loss Landscapes

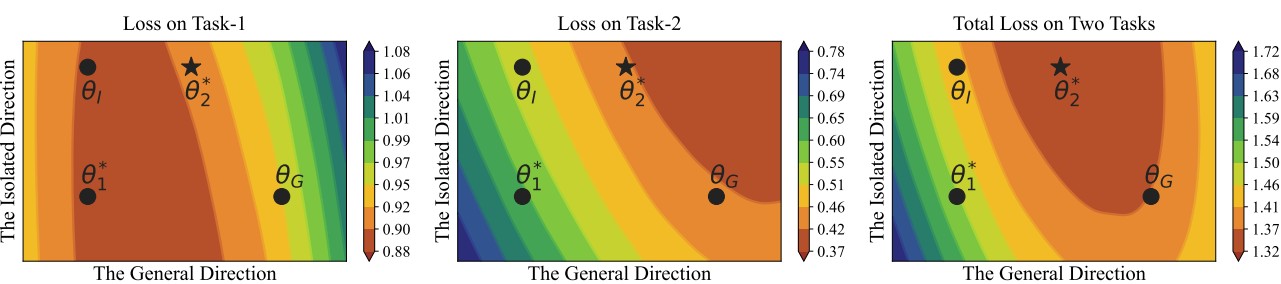

*Figure 11.* Visualization of the loss landscapes on the first task $\mathcal{D}^1$, the second task $\mathcal{D}^2$, and two tasks on 10S-ImageNetA.

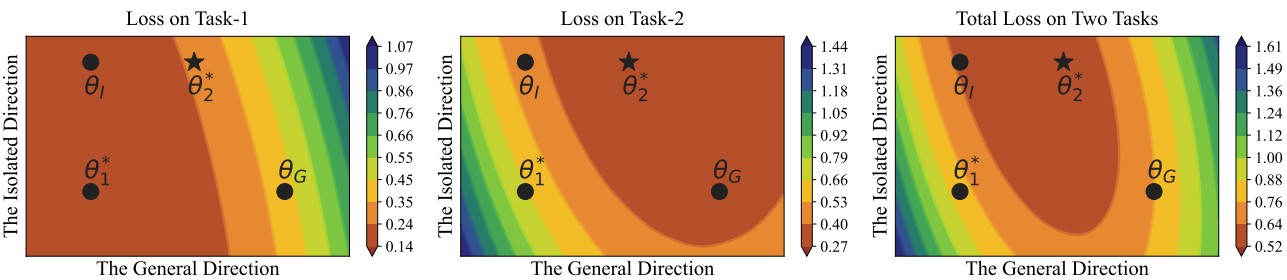

*Figure 12.* Visualization of the loss landscapes on the first task $\mathcal{D}^1$, the second task $\mathcal{D}^2$, and two tasks on 10S-ImageNetR.

To better understand how our two-branch LoRA updates enhances the stability-plasticity trade-off, we visualize the loss landscapes along the general directions (LoRA$_G$) and the isolated directions (LoRA$_I$) on 10S-ImageNetA and 10S-ImageNetR datasets. The results are shown in Fig.11 and Fig.12. Specifically, starting from the parameters $\theta_1^\star = \mathbf{W}^1$ after the first session, we construct a 2D plane spanned by the two update directions and evaluate the loss on a grid of coefficients $\alpha, \beta \in [-0.5, 1.3]$. Each point on the surface corresponds to $\theta(\alpha, \beta) = \theta_1^\star + \alpha w_G \mathbf{B}_G \mathbf{A}_G + \beta \mathbf{B}_I \mathbf{A}_I$. In Fig.11 and Fig.12, $\theta_G = \theta_1^\star + w_G \mathbf{B}_G \mathbf{A}_G$ and $\theta_I = \theta_1^\star + \mathbf{B}_I \mathbf{A}_I$. For simplicity, we set $\theta_2^\star = \theta_1^\star + \frac{1}{r} \sum_{j=1}^r \gamma^{(j)} w_G \mathbf{B}_G \mathbf{A}_G + \mathbf{B}_I \mathbf{A}_I$.

As shown in the results, the isolated direction lies in a region where the loss on $\mathcal{D}^1$ varies minimally while the loss on $\mathcal{D}^2$ changes obviously, indicating that it indeed captures truly task-specific knowledge that effectively adapts to the new task with limited interference to previous tasks. The general direction induces large loss variations on both tasks, reflecting its role as a shared subspace that captures general knowledge. Notably, a moderate update along the general direction for the new task does not induce loss increases on the old task. Instead, it moves the solution toward the center of the low-loss region of the old task, which suggests that it promotes positive cross-task knowledge transfer. Overall, the visualizations demonstrate that LoDA's effectiveness on finding a joint optimum during continual adaptation.

