# OpenReview forum: "Task-Driven Subspace Decomposition for Knowledge Sharing and Isolation in LoRA-based Continual Learning"
_ICML.cc/2026/Conference — ICML 2026 regular_

### Official Review · Reviewer_MaVH · 2026-02-24

**Soundness:** 3
**Presentation:** 3
**Significance:** 3
**Originality:** 3
**Overall Recommendation:** 5
**Confidence:** 4

**Summary:**

This paper proposes a novel variant of LoRA for class-incremental learning. It explicitly and data-drively optimizes the task-shared and task-specific spaces, supported by comprehensive experiments and theoretical proofs.

**Compliance With Llm Reviewing Policy:**

Affirmed.

**Final Justification:**

The paper is technically solid. The rebuttal addressed my concerns. I will keep my score.

**Key Questions For Authors:**

+ Can LoDA surpass baselines on larger backbones such as ViT-L/16, self-supervised variants such as DINOv2 as well as language and vision-language models?
+ Can you provide the GPU memory used during the training stage compared with baselines?

**Limitations:**

yes

**Strengths And Weaknesses:**

# Strength
+ comprehensive experiments and theoretical proofs
+ Compared to null-space approaches, explicitly solving for task-shared and task-specific spaces offers better plasticity, while also providing stronger theoretical guarantees than implicit optimization methods.

# Weakness
+ "Theorem 3.1 suggests that LoRA's learning capacity is determined by the magnitude of X's component in the subspace spanned by A." I argue that this conclusion does not hold. If it were true, then in non-continual learning scenarios, LoRA's A would no longer require training and could be determined simply by a single forward pass through the training set, which contradicts widely observed empirical results. However, this does not undermine the validity of the LoDA method itself, which explicitly solves for task-shared and task-specific spaces.
+ More memory used during the training stage compared with other LoRA-based method.
+ The meaning of the memory is unclear, which can be GPU memory usage or system memory usage. Both of them are needed to be provided.

---

> ### Author Rebuttal · Authors · 2026-03-31
>
> **Q1: The claim of Theorem 3.1.**
>
> A: Thanks for this insightful comment. We would like to clarify **the intended scope of Theorem 3.1**. Theorem 3.1 is derived under a conditional setting where the LoRA down-projection $A$ is fixed and only $B$ is updated. Therefore, its conclusion is not that $A$ is unnecessary to train in general LoRA, but that once $A$ is fixed, the effectiveness of learning $B$ is largely determined by the projection energy $\Vert XA\Vert_2^2$.
>
> **Why fixed $A$ setting is meaningful?** First, recent work [1] identifies the asymmetry of LoRA in **general non-CL tasks**, where freezing $A$ and tuning only $B$ can achieve performance close to standard LoRA. Second, prior CL methods such as InfLoRA suggest that fixing $A$ to span the null space of old-task data can mitigate interference between old and new tasks. **These works suggest**: (1) In CL, a properly fixed $A$ makes knowledge correlations and interference controllable while maintaining plasticity for learning new tasks. This is exactly the motivation behind our Theorem 3.1 and subspace decomposition; (2) Beyond CL, with a properly chosen subspace, a fixed $A$ may still support effective adaptation, although our primary focus is the CL setting.
>
> Overall, in this work, Theorem 3.1 mainly concerns how to design $A$ for CL in the fixed $A$ regime, while its underlying insight may also extend beyond CL.
>
> [1] Asymmetry in Low-Rank Adapters of Foundation Models (ICML 2024).
>
> **Q2: Experiments on other backbones.**
>
> A: **We evaluate LoDA on self-supervised backbones (DINO and iBOT), ViT-L/16 and vision-language model CLIP.** The results are in Table 1. For CLIP, we build upon the SimpleCIL baseline to leverage its zero-shot capability. For other backbones, we evaluate LoDA under the same pipeline as the main experiments. Overall, the results suggest that LoDA generalizes well across different pre-training paradigms and model scales. More detailed analysis can be found in Appx.C.3-C.5.
>
> Table 1
> |Backbone|Method|Last on 10S-INR|Avg on 10S-INR|
> |-|-|-|-
> DINO-1k|SeqLoRA|54.87|69.87
> -|InfLoRA|68.31|76.35
> -|VPT-NSP|68.96|76.22
> -|**LoDA(Ours)**|**73.66**|**80.51**
> iBOT-1k|SeqLoRA|56.28|71.15
> -|InfLoRA|71.84|78.20
> -|VPT-NSP|73.25|79.65
> -|**LoDA(Ours)**|**76.83**|**82.83**
> ViT-L/16|SeqLoRA|78.27|85.43
> -|**LoDA(Ours)**|**85.48**|**89.96**
> CLIP|SeqLoRA|78.45|84.32|
> -|RAPF|79.62|86.28
> -|SimpleCIL|74.48|81.06
> -|**LoDA(Ours)**|**82.65**|**87.36**
>
> **Q3: Analysis of memory usage.**
>
> A: Thanks for your valuable comment. We clarify that, **''Memory'' in Tab.6 and Appx.C.2 in our manuscript refers to auxiliary non-trainable storage maintained across tasks**, such as exemplars, prototypes, or statistics. Since our method does not introduce explicit additional system memory modules beyond standard training pipeline, we provide **an analysis of GPU memory and training cost in Table 2**. All results are measured on one NVIDIA RTX 4090 GPU with batch size 48. For the compared methods, we follow their official settings and set the batch size to 48.
>
> **Analysis of the overhead from GAO**.The main overhead of LoDA comes from GAO. **GAO decrease peak GPU memory**, because (1) it processes two smaller label-disjoint groups separately and the memory is determined by the larger ones rather than the full batch and (2) it does not keep extra optimizer states. **Compared to SGD/Adam optimizer**, since GAO computes inner gradients, it inevitably requires $2\times$ forward & backward passes, which leads to **about $2\times$ GFLOPs and training time**. **While compared to perturbation- (e.g.,SAM[2]) or meta-learning-based methods**, such overhead is comparable.
>
> **Comparison with existing method.** We compare LoDA with representative VPT-based method VPT-NSP and LoRA-based method InfLoRA. LoDA incurs extra overhead due to GAO, but **GAO should be viewed as an auxiliary regularizer rather than the core working mechanism of LoDA**. Without GAO, LoDA already achieves strong performance while maintaining comparable training cost and storage to existing methods. We also implement an efficient variant of LoDA with LoRA rank 4, which introduces 0.22M extra parameters (0.25% of the backbone), yet still outperforms existing methods.
>
> [2] Sharpness-aware minimization for efficiently improving generalization (ICLR 2021).
>
> Table 2
> |Method|Training|||||Inference|||
> |-|-|-|-|-|-|-|-|-|
> ||GPU Memory (MiB)|GFLOPs/batch|Time/batch (s)|Storage(MB)|Extra Params|GPU Memory (MiB)|Extra Params|Last on 10S-INR
> SeqLoRA|4560|867|0.12s|0.0|1.77M(2.06%)|1072|0.0M|72.09
> InfLoRA(CVPR-24)|4465|848|0.12s|11.3|0.37M(0.43%)|1072|0.0M|75.06
> VPT-NSP(NIPS-24)|4474|851|0.12s|27.0|0.09M(0.10%)|1080|0.09M|78.67
> **LoDA w/o GAO**|4560|867|0.12s|27.0|1.77M(2.06%)|1072|0.0M|81.05
> **LoDA w/ SAM**|4560|1734|0.25s|27.0|1.77M(2.06%)|1072|0.0M|80.77
> **LoDA**|3686|1734|0.26s|27.0|1.77M(2.06%)|1072|0.0M|81.93
> **LoDA (r=4)**|3592|1686|0.12s|27.0|0.22M(0.25%)|1072|0.0M|81.26
>
> We will add these analysis in final version.

---

> > ### Author Rebuttal · Reviewer_MaVH · 2026-04-02
> >
> > The rebuttal addressed my concerns. I will keep my score.

---

### Official Review · Reviewer_svmg · 2026-03-05

**Soundness:** 2
**Presentation:** 3
**Significance:** 3
**Originality:** 2
**Overall Recommendation:** 3
**Confidence:** 4

**Summary:**

This paper studies LoRA-based continual learning under class-incremental settings. It argues prior “null-space” style isolation can miss shared directions and can also be ineffective when tasks are correlated. The method, LoDA, decomposes the LoRA update space into a general subspace (high energy for old+new data) and an isolated subspace (high new/old energy ratio), freezes the down-projections on these bases, trains up-projections with a gradient-alignment objective (GAO), and applies a closed-form rescaling to recalibrate the general update before merging. Experiments on several vision CL benchmarks show consistent gains over recent LoRA/PEFT baselines.

**Compliance With Llm Reviewing Policy:**

Affirmed.

**Final Justification:**

The rebuttal addresses some practical concerns, but my main issues remain only partially resolved. In particular, the support for GAO is still not strong enough in the correlated-but-conflicting regime, and the added TIL/DIL results are not yet sufficient to establish broader generality beyond the current ViT-based vision CIL setting. Therefore, I keep my weak reject recommendation.

**Key Questions For Authors:**

1. How sensitive is LoDA to rank choice and to the weights w_G and \lambda? Please show a small sensitivity sweep.
2. Can you test on a non-class-incremental CL setting (e.g., domain-incremental or task-incremental) to show the idea is general?
3. What is the compute overhead of GAO compared to standard training? Please report wall-clock or step-time.
4. The isolated subspace assumes past statistics are full-rank / well-behaved. What happens when this is not true (small data, heavy redundancy)?

**Limitations:**

They should more directly discuss: (i) dependence on data statistics quality, (ii) extra compute from GAO, and (iii) limited evidence beyond ViT vision class-incremental benchmarks.

**Strengths And Weaknesses:**

Strengths
1. The core idea is easy to grasp: split LoRA updates into shared and isolated parts.
2. The objectives are simple and mostly well-motivated by the projection-energy view.
3. The training pipeline is consistent with the design goal of reducing interference.
4. The experiments show improvements over several LoRA/PEFT continual learning baselines.

Weaknesses
1. The theoretical analysis feels limited. It mainly supports a simplified setting and does not reflect the full training dynamics.
2. The method depends on estimating subspaces from data statistics. This may be unstable when old-task data is small or highly correlated.
3. The isolated-subspace construction relies on matrix properties that may not hold well in practice. When rank is low, the split may be noisy.
4. GAO is introduced as a key stabilizer, but the paper does not fully explain when it helps most and when it might hurt.
5. The system has multiple components and hyperparameters. It is hard to tell how sensitive the final gains are without broader sweeps.
6. Some baselines may not be tuned equally. Small tuning differences can matter a lot in continual learning.
7. Most results are on vision class-incremental benchmarks with ViT. It is unclear if the same idea works for other CL setups.
8. The paper focuses on average accuracy, but it gives less detail on trade-offs like memory, speed, and per-task overhead.

---

> ### Author Rebuttal · Authors · 2026-03-31
>
> **Q1: Hyper-parameter analysis.**
>
> A: We study how hyper-parameters affect performance on ImageNetR(INR) and ImageNetA(INA), including LoRA rank, $w_G$ and $\lambda$. Results are in Table 1&2&3. We thus set rank to 16, $w_G$=0.5 and $\lambda$=3.0. More analysis is in Appx.C.1&C.2.
>
> Table 1
> |rank|Acc(20S-INR)|Extra Params|
> |-|-|-|
> |4|78.1|0.2M(0.3%)|
> |8|78.5|0.4M(0.5%)|
> |16|79.1|0.9M(1.0%)|
> |64|78.9|3.5M(4.1%)|
>
> Table 2
> |$w_G$|Acc(10S-INA)|Acc(10S-INR)|
> |-|-|-|
> |0.1|60.19|80.48|
> |0.3|62.52|81.68|
> |0.5|62.59|81.93|
> |0.7|61.60|81.42|
>
> Table 3
> |$\lambda$|Acc(10S-INA)|Acc(10S-INR)|
> |-|-|-|
> |1.0|61.59|81.49|
> |3.0|62.59|81.93|
> |10.0|62.45|81.52|
> |15.0|62.04|81.21|
>
> **Q2: Experiments on non-CIL settings.**
>
> A: To show LoDA’s generality beyond CIL, we evaluate it on TIL and DIL: TIL on 10S-INA and 20S-INR(Table 4), and DIL on Office-Home(Table 5), and show that it generalizes well on TIL and DIL. Unlike methods tailored to classifier design, LoDA is a more general parameter-level framework.
>
> Table 4(TIL)
> |Method|10S-INA||20S-INR||
> |-|-|-|-|-|
> ||Acc|BWT|Acc|BWT|
> |SeqLoRA|83.5|-4.31|88.4|-7.63|
> |InfLoRA|83.4|-2.54|93.2|-2.83|
> |LoDA (Ours)|85.7|-1.23|96.0|-1.20|
>
> Table 5(DIL)
> ||Last(Office-Home)|Avg(Office-Home)|
> |-|-|-|
> |L2P|80.03|79.72|
> |RanPAC|82.28|82.30|
> |DCE|84.50|86.40|
> |DUCT|86.91|86.27|
> |LoDA (Ours)|87.09|88.93|
>
> **Q3: Overhead of GAO.**
>
> A: Table 6 reports GPU memory, computational cost, and batch time on 1 NVIDIA RTX 4090 with batch size 48. **GAO decrease GPU memory** because each forward pass uses a smaller group and it does not keep extra optimizer states. **Compared with SGD/Adam optimizer**, since GAO computes inner gradients, it requires two forward&backward passes, leading to about 2× GFLOPs and time. **While compared with widely used perturbation-based optimizer (e.g., SAM)**, this overhead is comparable.
>
> Table 6
> ||GPU Memory(MiB)|GFLOPs/batch|Time/batch(s)|Acc(10S-INA)
> |-|-|-|-|-
> SeqLoRA|4560|867|0.12s|53.24
> LoDA w/o GAO|4560|867|0.12s|61.27
> LoDA w/ SAM|4560|1734|0.25s|61.02
> LoDA|3686|1734|0.26s|62.59
>
> **Q4: Full-rank/stability concerns.**
>
> A: The full-rank assumption generally holds in standard real-world datasets. For ViT-B/16, $X\in\mathbb{R}^{(197×N)×768}$. Even with 16-shot data, each class gives $197×16=3152$ patch tokens for a 768-dimensional space. Although some work treats token features as approximately low-rank by ignoring small singular values, their variations across spatial locations and images still make $S=X^\top X$ **full-rank in practice without affecting our isolated solver**. As shown by the 16-shot INR (13% data) ablation(Table 7), the decomposition remains effective in the low-data regime. Moreover, the isolated subspace solver can also use a small ridge term, $S=S+\epsilon I$, to ensure stability in ill-conditioned cases.
>
> Table 7
> ||Acc, 10S-INR(16-shot)|Acc, 20S-INR(16-shot)|
> |-|-|-|
> SeqLoRA|64.7|61.5
> w/o $LoRA_G$|68.2|67.5
> w/o $LoRA_I$|68.6|66.4
> LoDA|70.2|69.8
>
> **Q5: When GAO helps and when it might hurt.**
>
> A: **Working mechanism**. In Appx.B.4, we show GAO's objective can be approximated as $L_{GAO} \approx L_{CE}-\rho g_1^\top g_2(\frac{1}{\Vert g_1\Vert_2^2}+\frac{1}{\Vert g_2\Vert_2^2})$, which implicitly aligns intra-task cross-group gradients. We also show that this alignment **reduce loss increase caused by future interference**.
>
> We verify this by 3 task-wise indicators on 10S-INA(Table 8): (1)Conflict Ratio(CR): we compute gradients $g_1,g_2$ of two intra-task groups and count a conflict when $cos(g_1,g_2)<-\epsilon$; (2)$cos(W_o$,$W_n)$: cosine similarity between the old weight $W_o=\sum_{i=1}^{t-1}B_iA_i$ and the new weight $W_n=\sum_{i=1}^{t}B_iA_i$; (3)forgetting. Table 8 shows that GAO **reduces intra-task gradient conflicts** (lower CR), which in turn **reduces cross-task weight interference** (higher $cos(W_o,W_n)$), thus mitigates forgetting.
>
> **When it helps.** GAO is helpful when new and old tasks are correlated but their gradients conflict, since GAO suppresses potential conflicting components. **When it fails.** When new gradients are nearly orthogonal to the old space, the gradients aligned in old tasks are not reusable for the new task, so GAO reduces to standard training **without harming new-task learning**.
>
> Table 8
> ||CR (%, $\epsilon$=0.1)|$cos(W_o$,$W_n)$|Forgetting
> |-|-|-|-|
> w/o GAO|5.59|0.82|14.9
> w/ GAO|1.86|0.87|13.9
>
> **Q6: The theory considers a simplified setting.**
>
> A: Theorem 3.1 gives a general theory of controlling LoRA via fixing A. Based on this, our subspace decomposition specifies how A should be designed under old–new task interactions. Hence, our theory is developed for knowledge correlation and interference dynamics in CL. We will futher clarify it in final version.
>
> **Q7: Some baselines may not be tuned equally.**
>
> A: SOTAs are evaluated using their official settings and all ablations are conducted under the same fixed setting. We believe our comparison is fair and we have provided our code in the supplementary material.

---

> > ### Author Rebuttal · Reviewer_svmg · 2026-04-03
> >
> > On GAO theoretical grounding: The analysis in the rebuttal essentially says GAO won't hurt when tasks are nearly orthogonal, which sidesteps the harder question of correlated-but-conflicting scenarios that are actually common in CIL. The theoretical support remains shallow. Also on non-CIL generalization: The TIL/DIL experiments are appreciated but feel rushed, baselines are limited and the experimental setup is not well described. This doesn't convincingly demonstrate that the core subspace decomposition idea transfers beyond the standard CIL setting. Overall, while the authors made a reasonable effort, the responses to the more fundamental concerns feel more like damage control than genuine resolution. I'll keep my score at 3.

---

> > > ### Author Response · Authors · 2026-04-06
> > >
> > > Thanks for your comments that improve the readability of our paper.
> > >
> > > **Q1: On GAO theoretical grounding.**
> > >
> > > We sincerely thank the reviewer for the thoughtful comments. To the best of our understanding, we believe that the reviewer’s major concern is **the theoretical support for GAO in the correlated-yet-conflicting regime.** We would like to respectfully clarify that, in our rebuttal, **we have that GAO is helpful when old and new tasks remain correlated while exhibiting gradient conflict.** Our theory in **Appx.B.4** have covered this regime. In Eq.(B.4.7), we denote the old-task gradient as $g_{old}=\sum_{i=1}^{N}\gamma_i g_i$ with $\gamma_i\ge 0$, and model a future gradient as:
> > >
> > > $g_{fut}=\sum_{i=1}^{N}\beta_i g_i+\xi,\quad \beta_i\ge 0,\ \|\xi\|_2\le \epsilon,$
> > >
> > > where $\{g_i\}$ are gradients of groups from the old task. This captures **correlation** because **the main component of $g_{fut}$ (the first term) lies in the cone spanned by old gradients**. It also models **conflict**, since $g_{old}$ and $g_{fut}$ use different coefficients $\{\gamma_i\}$ and $\{\beta_i\}$ and thus can **prefer different directions**. Their inner product is $g_{old}^\top g_{fut}=\sum_{i=1}^{N}\sum_{j=1}^{N}\gamma_i\beta_j\, g_i^\top g_j+\sum_{i=1}^{N}\gamma_i g_i^\top \xi$, so conflict arises when $g_i^\top g_j$ are unfavorable.
> > >
> > > Eq.(B.4.9) further gives the old loss change after a future update:
> > >
> > > $\mathcal{L}(\mathcal{D},\theta^+)-\mathcal{L}(\mathcal{D},\theta)
> > > \le
> > > -\eta \sum_{i=1}^{N}\sum_{j=1}^{N}\gamma_i\beta_j\, g_i^\top g_j
> > > -\eta \sum_{i=1}^{N}\gamma_i g_i^\top \xi
> > > +\mathcal{O}(\eta^2\|g_{fut}\|_2^2).$
> > >
> > > Therefore, when new and old tasks are correlated($\beta_j>0$), the interference of future update is governed by $g_i^\top g_j$. In this regime, GAO promotes larger $g_i^\top g_j$, thereby tightening the upper bound on old loss and reducing forgetting. By contrast, when the new task is nearly orthogonal to the old subspace ($\beta_j\approx 0$), GAO reduces to standard training.
> > >
> > > Our experiments support this analysis. On 10S-ImageNetA(Tab.8 in rebuttal), $\cos(W_o,W_n)$ is consistently high (>0.8), suggesting that correlated old and new updates are common in CIL. In this regime, GAO reduces forgetting(Tab.8 in rebuttal) and brings stable gains in ablation study(Tab.3 in the manuscript). We believe these results are consistent with our theoretical analysis.
> > >
> > > **Q2: The TIL/DIL experiments feel rushed.**
> > >
> > > A: **Baselines.** We would like to clarify that, **due to the rebuttal limit(5000 characters)**, our non-CIL experiments in the rebuttal **focused on the strongest and most relevant baselines**. For DIL, we compare against DCE(ICML-25) and DUCT(CVPR-25), which are to the best of our knowledge the latest and strongest **published methods**. For TIL, we choose InfLoRA because its null-space LoRA design is relevant to our motivation. To provide a comprehensive comparison, we reproduced and added several representative baselines, as shown in Table 1 and Table 2.
> > >
> > > **Experimental setups.** We clarify that our experimental setup is kept consistent with prior methods, **ensuring a completely fair comparison**. For TIL, the setup follows Implementation Details, and task-ID is available during inference. For DIL, we use OfficeHome with 4 domains (Product, Clipart, Real_World, Art). Following existing works [1], we adopt the ViT-B/16 as our backbone, the training epoch is set to 15, the batch size is 128 and the initial learning rate is 0.1. All other hyperparameters are kept unchanged.
> > >
> > > **Evaluation metrics. For fair comparison**, in **DIL**, we report $A_B$ and $\overline{A}$ following [1], where $A_B$ represents the final-session accuracy and $\overline{A}$ represents the average accuracy over all sessions. In **TIL**, we report **Acc** and **BWT** following [2], where Acc is the overall accuracy and BWT reflects the forgetting.
> > >
> > > [1]Dual Consolidation for Pre-Trained Model-Based Domain-Incremental Learning.
> > > [2]TRGP: Trust Region Gradient Projection for Continual Learning.
> > >
> > > Table 1(DIL)
> > > ||Venue|$A_B$(Office-Home)|$\overline{A}$(Office-Home)|
> > > |-|-|-|-|
> > > |L2P|CVPR-22|80.03|79.72|
> > > |S-iPrompt|NIPS-22|80.51|81.50|
> > > |CODA-Prompt|CVPR-23|85.07|84.70|
> > > |RanPAC|NIPS-23|82.28|82.30|
> > > |EASE|CVPR-24|76.33|81.16|
> > > |DCE|ICML-25|84.50|86.40|
> > > |DUCT|CVPR-25|86.91|86.27|
> > > |**LoDA (Ours)**|-|**87.09**|**88.93**|
> > >
> > > Table 2(TIL)
> > > |Method|Venue|10S-ImageNetA||20S-ImageNetR||
> > > |-|-|-|-|-|-|
> > > |||Acc|BWT|Acc|BWT|
> > > |SeqLoRA|-|83.5|-4.31|88.4|-7.63|
> > > |InfLoRA|CVPR-24|83.4|-2.54|93.2|-2.83|
> > > |SD-LoRA|ICLR-25|83.9|-2.75|94.6|-2.62|
> > > |MACIL|ICML-25|85.1|-1.58|95.5|-1.36|
> > > |PLAN|ICCV-25|83.1|-2.87|93.4|-2.79
> > > |**LoDA(Ours)**|-|**85.7**|**-1.23**|**96.0**|**-1.20**|
> > >
> > > # We therefore kindly ask the reviewer to reconsider the score in light of our clarifications and the additional evidence in the rebuttal.

---

### Official Review · Reviewer_SrwD · 2026-03-09

**Soundness:** 3
**Presentation:** 2
**Significance:** 3
**Originality:** 2
**Overall Recommendation:** 3
**Confidence:** 4

**Summary:**

This paper presents a LoRA-based continual learning method that addresses the stability-plasticity dilemma through task-driven subspace decomposition. The proposed method decompose the update space into general and isolated subspaces by maximizing projection energy objectives. LoDA constructs general subspaces via joint energy maximization and isolated subspaces through relative energy ratios, ensuring effective knowledge sharing and isolation. LoDA introduces Gradient-Aligned Optimization for robust up-projection learning and derives closed-form recalibration for the general branch to approximate feature-level joint optima. Extensive experiments demonstrate state-of-the-art performance across multiple benchmarks.

**Compliance With Llm Reviewing Policy:**

Affirmed.

**Final Justification:**

The paper presents a technically interesting method with solid empirical gains and a clearer analysis of gradient alignment, efficiency, and ablations. I appreciate that the rebuttal addresses several practical concerns, such as storage overhead, SVD complexity, and comparisons with alternative subspace constructions, which improves confidence in the method’s efficiency. However, my main concerns regarding the generality of the assumptions, robustness across more diverse and realistic task settings, and the strength of the theoretical justification are only partially addressed. The empirical evaluation remains somewhat limited in scope, and it is still unclear how well the approach would generalize beyond the presented scenarios. Overall, while the rebuttal improves clarity, it does not substantially change my assessment, and I recommend a weak reject.

**Key Questions For Authors:**

1.Can you provide more direct evidence that gradient alignment indeed reduces forgetting and/or improves generalization (e.g., correlation analysis between alignment metrics such as cosine similarity / conflict rate and forgetting; layer-wise or task-wise alignment trajectories)?

2.For the O(D^3)per-layer SVD, do you use truncated / randomized SVD, low-rank approximations, or blockwise strategies? If so, what is the accuracy–efficiency trade-off?

3.Under what assumptions does the closed-form rescaling in Eq. (13) hold (e.g., approximating a feature-level joint optimum)? How stable is it when tasks are highly dissimilar or when feature drift is strong?

**Limitations:**

1.The current justification for Gradient-Aligned Optimization is largely intuitive and empirically motivated. More theoretical grounding or diagnostic experiments are needed to demonstrate robustness across varying task difficulties and conflict levels.

2.Since the approach relies on energy decomposition to separate shared vs. isolated directions, the optimal sharing/isolation ratio may vary with task relatedness. Suboptimal subspace dimensionality or thresholds could lead to negative transfer (over-isolation) or increased forgetting (over-sharing).

**Strengths And Weaknesses:**

Strengths:

1.Unlike prior works that rely on null-space approximation, LoDA introduces a relative energy maximization objective that explicitly optimizes directions strongly activated by new tasks but weakly activated by previous ones. This formulation provides a more principled mechanism for disentangling task-specific knowledge. Moreover, the Cholesky factorization trick ensures computational tractability and numerical stability during optimization.

2.The paper provides analytical insights into subspace decomposition. In particular, the rescaling factor in Eq. (13) yields an exact solution for minimizing the joint feature optimization error, thereby avoiding the local linearity assumptions.

3.The experimental evaluation is comprehensive. Both ablation studies and hyperparameter sensitivity analyses are thoroughly conducted and clearly presented.

Weaknesses:

1.The paper lacks a rigorous analysis of whether gradient alignment theoretically improves generalization performance or effectively mitigates catastrophic forgetting. A deeper theoretical or empirical investigation would strengthen the claims.

2.Although the paper claims scalability, storing $S^{(1:t-1)} \in \mathbb{R}^{L \times D \times D} $ for all layers (approximately 27MB for ViT-B/16) may grow in practice due to numerical stability requirements for Cholesky updates across long task sequences. While the memory cost is constant per task, the computation of the SVD on $S^{(1:t-1)} + S^t$ scales as $O(D^3)$ per layer, which could become prohibitive for larger embedding dimensions.

3.Since the method is built upon energy decomposition, it would be more convincing to include comparisons with alternative subspace-based approaches, such as PCA-based decomposition, Fisher information subspace methods, and Gradient Projection Memory (GPM)-style subspace learning.

4.An additional ablation study without the recalibration step is recommended to explicitly validate the contribution of the closed-form merging strategy.

---

> ### Author Rebuttal · Authors · 2026-03-31
>
> **Q1: Analysis of GAO.**
>
> A: In Appx.B.4, we show that GAO implicitly **align intra-task cross-group gradients**: it can be approximated as $L_{GAO} \approx L_{CE}-\rho g_1^\top g_2(\frac{1}{\Vert g_1\Vert_2^2}+\frac{1}{\Vert g_2\Vert_2^2})$. We also show that improving such alignment **reduces old-task loss increase caused by future interference** under correlated tasks.
>
> We analyze **4 task-wise indicators** on 10S-ImageNetA (Table 1):
> (1)Conflict Ratio(CR): we compute gradients $g_1,g_2$ of two intra-task groups and count a conflict when $cos(g_1,g_2)< -\epsilon$;
> (2)$cos(W_o$,$W_n)$: similarity between the old weight $W_o=\sum_{i=1}^{t-1}B_iA_i$ and the new weight $W_n=\sum_{i=1}^{t}B_iA_i$;
> (3)old-task loss $L_o$;
> (4)forgetting.
>
> Table 1 show that GAO **reduces intra-task conflicts** (lower CR), encouraging learning robust and transferable updates. This leads to **less interference between learned weights across tasks** (Increased $cos(W_o$,$W_n)$), thereby **improving old-knowledge retention** (lower $L_o$ and forgetting). It is consistent with our theoretical analysis.
>
> Table 1
> |Task-Idx||Avg|1|2|3|4|5|6|7|8|9
> |-|-|-|-|-|-|-|-|-|-|-|-
> |CR (%, $\epsilon$=0.1)|w/o GAO|5.59|7.7|3.8|2.5|4.2|7.3|8.5|4.5|8.0|3.8
> ||w/ GAO|1.86|2.3|1.7|0.4|2.1|1.2|3.5|0.9|3.0|1.6
> |$cos(W_o$,$W_n)$|w/o GAO|0.82|0.63|0.74|0.82|0.82|0.84|0.89|0.9|0.87|0.86
> ||w/GAO|0.87|0.66|0.82|0.86|0.89|0.9|0.93|0.94|0.91|0.91
> |$L_o$|w/o GAO|1.50|0.91|1.13|1.25|1.41|1.49|1.70|1.78|1.85|1.94
> ||w/ GAO|1.36|0.85|1.04|1.14|1.27|1.37|1.54|1.6|1.67|1.74
> |Forgetting|w/o GAO|14.9|18.3|12.9|15.4|16|13.3|15.4|14.4|13.7|15.0
> ||w/ GAO|13.9|18.3|10.9|14.0|15.5|12.8|14.1|12.8|12.9|14.2
>
> **Q2: Overhead of storing $S$ & SVD complexity.**
>
> A: **Storage.** LoDA stores one $D^2$ matrix $S$ per layer. It is scalable because (1)**this cost is independent of task number** and (2)**$S$ is a non-trainable buffer rather than params** so it induces no optimizer states. For a ViT with $L$ layers and width $D$, the total storage is $LD^2$. In comparison, each ViT block contains about $12D^2$ params per layer ($4D^2$ from attention and $8D^2$ from MLP), so the relative storage is $LD^2/12LD^2 \approx 8.3\%$. Due to approximation error, we report the empirical storage(Table 2). **Overall, the relative cost does not grow with $D$ and is small vs. backbone.** Notably, methods like Adam-NS & VPT-NSP also store a $D^2$ matrix per layer, yet LoDA performs better(Tab.6 in our manuscript).
>
> Table 2
> ||$D$|ViT|$S$
> |-|-|-|-
> |ViT-B/16|768|330MB|27MB(8.2%)
> |ViT-L/16|1024|1161MB|96MB(8.3%)
>
> **Complexity of SVD.** We use full SVD in LoDA. In Table 3, we use Randomized SVD with projection dimension $q$ to reduce the complexity from $O(D^3)$ to $O(qD^2)$ while maintaining performance. Since subspace decomposition (the only step involving SVD) is operated only once before each task, **its time (0.24s per layer) is negligible vs. training (54s per task on 10S-INA)**.
>
> Table 3
> ||Complexity|Time(s)|Acc(10S-INA)|
> -|-|-|-
> Rand(q=64)|$O(qD^2)$|0.05|62.3
> Rand(q=128)|$O(qD^2)$|0.08|62.2
> Full SVD|$O(D^3)$|0.24|62.6
>
> **Q3: Compared with other subspace construction methods.**
>
> A: Table 4 compare LoDA with:(1)GPM & DualGPM, since LoDA does not compute null space, we use **InfLoRA’s subspace construction w/ GPM/DualGPM-based old null space in our framework**; (2)a PCA-style variant LoDA(PCA), by replacing $\Vert XU\Vert_2^2$ with $\Vert (X-\mu)U\Vert_2^2$.
>
> Table 4
> ||Acc(10S-INA)|
> |-|-|
> |GPM+InfLoRA|56.8|
> |DualGPM+InfLoRA|57.1|
> |LoDA(PCA)|61.3|
> |LoDA(Ours)|62.6|
>
> **Q4: Ablation of the recalibration.**
>
> A: In Table 5, w/o recalibration, LoDA largely reduces to sequential tuning since $LoRA_G$ and $LoRA_I$ together cover the full update directions of the new task. LoDA's working mechanism is to decompose them for separate control (recalibration for $LoRA_G$ and direct adding for $LoRA_I$).
>
> Table 5.
> ||Acc(10S-INA)|Acc(10S-INR)|
> |-|-|-|
> |SeqLoRA|53.2|68.3|
> |LoDA w/o recalibration|57.8|74.4|
> |LoDA|62.6|79.0|
>
> **Q5: Assumption & stability of rescaling(Eq.13).**
>
> A: Eq.13 solves Eq.12 w/o assumptions. Eq.12 approximates a feature-level joint optimum on tasks $1:t$ by minimizing feature optimization error, assuming (1) $W^{t-1}$ is a good joint weight for tasks $1:t-1$, and (2) $w_GB_GA_G$ is the optimal general update for task $t$. Thus, minimizing drift of old&new tasks from their desirable weights seeks a joint optimum. Since a mild step toward $W^t$ can preserve or even improve old loss (Fig.4), old-task constraints can be relaxed by setting $\lambda>1$.
>
> To test its stability under dissimilar tasks, we build **a 2-task CL setting with two dissimilar fine-grained datasets, Aircraft(w/ planes) and CUB(w/ birds) (30 classes each)**, and sweep $\lambda$ from $0$ (no adaptation to new task in $LoRA_G$) to $\infty$ (fully new weights). Table 6 show that it consistently finds a weight with lower joint loss.
>
> Table 6
> |$\lambda$|Acc|
> |-|-|
> |0.0|78.92|
> |1.0|81.40|
> |3.0|82.05|
> |10.0|81.68|
> |+$\infty$|80.50|

---

> > ### Author Rebuttal · Reviewer_SrwD · 2026-04-03
> >
> > The authors have addressed some of my concerns (e.g., analysis and efficiency), but key issues regarding the generality of assumptions and robustness under broader task settings remain insufficiently resolved, so I recommend a weak reject.

---

> > > ### Author Response · Authors · 2026-04-06
> > >
> > > Thanks for your valuable comments, which help improve the quality of our paper.
> > >
> > > **Q1:Generality of assumption.**
> > >
> > > We sincerely thank Reviewer SrwD for the constructive feedback. We would like to respectfully clarify that our core theory focuses on the analysis of the LoRA $A$ and subspace decomposition is developed under relatively mild assumptions.To the best of our understanding, these derivations and assumptions do not appear to be the main point of concern. Therefore, **we believe Reviewer SrwD's major concern is whether our rescaling is general enough to approximate a feature-level joint optimum under varying task similarity**.
> > >
> > > **Theory.** We substitute the solution in Eq.13 into the objective $L(B_G^{(j)\star})$ in Eq.12:
> > >
> > > $L(B_G^{(j)\star})=\frac{\lambda \left(A_G^{(j)} S^t A_G^{(j)\top}\right)\left(A_G^{(j)} S^{1:t-1} A_G^{(j)\top}\right)}{A_G^{(j)} \left(\lambda S^t + S^{1:t-1}\right) A_G^{(j)\top}}\left\| B_G^{(j)}\right\|_F^2,$
> > >
> > > and we have:
> > >
> > > $L(B_G^{(j)\star})\le L(0), \quad L(B_G^{(j)\star})\le L(B_G^{(j)}).$
> > >
> > > Here, we omit the derivation. It shows that the rescaling is provably better than both using fully new weight ($L(B_G^{(j)})$) and fully old weight ($L(0)$), **supporting that it steer model toward a joint lower-loss region. This theory is general and does not rely on assumptions on task similarities**.
> > >
> > > **Empirical study. We build 2 benchmarks:**
> > >
> > > (1)We build a **5-task class incremental learning(CIL) benchmark** by combining Aircraft, Cars, CUB, ImageNet-A, and UCF datasets. **It explicitly cover different task similarities**: Cars and Aircraft are closer due to their focus on man-made objects; CUB and ImageNet-A are closer as both involve animal images; UCF shows the largest shift with action categories. We sample 100 classes from each dataset to ease data split. We report the task-by-task CIL & TIL acc. (Table 1 & Table 2).
> > >
> > > (2)We use **the domain-incremental learning(DIL) benchmark Office-Home** with 4 domains(Product,Real_World,Art,Clipart, denoted as P,R,A,C), **which also exhibits varying task similarities**: Product and Real_World are closer because both preserve realistic appearance; Art and Clipart show larger shifts due to their distinct textures. We report acc. of each domain and the average acc. in Table 3.
> > >
> > > **Analysis.** We vary $\lambda$ from 0.0 to $\infty$, which **moves from using no new weight to fully using it in $LoRA_G$**. Across all stages, rescaling with $1.0\le\lambda \le10.0$ show stable improvements over the two extremes($\lambda=0.0$ and $\lambda=\infty$). **Such stable gains suggest that it is effective across varying task similarities, which is consistent with our theory.**
> > >
> > > Table 1(CIL)
> > > |Task-Idx|1|2|3|4|5(Last)|Avg|
> > > |-|-|-|-|-|-|-|
> > > $\lambda=0.0$(w/o new weight)|80.35|74.25|72.55|73.67|72.67|74.70
> > > $\lambda=1.0$|80.35|76.25|74.43|76.34|73.50|76.17
> > > $\lambda=3.0$|80.35|75.53|74.42|76.39|**74.58**|**76.25**
> > > $\lambda=10.0$|80.35|74.17|72.42|74.48|74.04|75.09
> > > $\lambda=\infty$(fully new weight w/o rescaling)|80.35|71.54|68.68|70.77|72.12|72.69
> > >
> > > Table 3(TIL)
> > > |Task-Idx|1|2|3|4|5(Last)|Avg|
> > > |-|-|-|-|-|-|-|
> > > $\lambda=0.0$|80.35|81.07|79.71|82.19|82.73|81.21
> > > $\lambda=1.0$|80.35|81.56|81.07|83.56|84.00|82.11|
> > > $\lambda=3.0$|80.35|81.31|81.22|83.67|**84.19**|**82.15**
> > > $\lambda=10.0$|80.35|81.14|80.65|82.64|83.63|81.68
> > > $\lambda=\infty$|80.35|80.87|79.32|81.45|81.78|80.75
> > >
> > > Table 3(DIL)
> > > |Domain|P|C|R|A|Avg
> > > |-|-|-|-|-|-
> > > $\lambda=0.0$|93.77|66.18|92.13|86.15|84.40
> > > $\lambda=1.0$|93.99|72.06|92.43|87.24|86.36
> > > $\lambda=3.0$|94.22|74.27|92.58|87.24|87.09
> > > $\lambda=10.0$|94.37|74.50|92.43|87.65|**87.22**
> > > $\lambda=\infty$|92.65|72.58|92.28|87.52|86.26
> > >
> > > **Q2:Robustness under border task settings.**
> > >
> > > A:We conduct ablation studies on the **5-task CIL benchmark and the Office-Home DIL benchmark described in our response to Q1. They capture different types of shifts and form border task settings in CL:** (1)CIL emphasizes semantic shifts, while DIL focuses on domain distribution shifts; (2)both benchmarks cover varying task similarities as discussed above. Table 4 & Table 5 show that our subspace decomposition and GAO yield consistent improvements.
> > >
> > > Table 4(CIL)
> > > |Task-Idx|1|2|3|4|5(Last)|Avg|
> > > |-|-|-|-|-|-|-|
> > > Seq-LoRA|80.20|61.28|56.94|55.68|61.96|63.21
> > > $LoRA_G$|80.20|72.21|70.99|71.54|70.73|73.13
> > > $LoRA_I$|80.20|72.48|71.33|72.64|71.51|73.63
> > > $LoRA_G$+$LoRA_I$ (LoDA w/o GAO)|80.20|73.99|72.68|74.34|72.97|74.84
> > > LoDA(Ours)|80.35|75.53|74.42|76.39|**74.58**|**76.25**
> > >
> > > Table 5(DIL)
> > > |Domain|P|C|R|A|Avg
> > > |-|-|-|-|-|-|
> > > Seq-LoRA|93.12|66.34|90.35|86.42|84.06
> > > $LoRA_G$|93.32|70.23|91.67|86.83|85.51
> > > $LoRA_I$|93.84|72.01|92.12|86.52|86.12
> > > $LoRA_G$+$LoRA_I$ (LoDA w/o GAO)|93.84|72.44|92.43|87.38|86.52
> > > LoDA(Ours)|94.22|74.27|92.58|87.24|**87.09**
> > >
> > > # We sincerely hope the reviewer will reconsider the score based on our clarifications and additional evidences.

---

### Official Review · Reviewer_PEpP · 2026-03-12

**Soundness:** 3
**Presentation:** 3
**Significance:** 2
**Originality:** 3
**Overall Recommendation:** 5
**Confidence:** 4

**Summary:**

This paper investigates LoRA-based CL and points out that the learning performance of LoRA is limited by the projection energy of task features onto the LoRA down-projection's row subspace. Based on this, the authors propose LoDA: a two-branch LoRA where the down-projection is fixed in two subspaces: (i) a general subspace found by maximizing the total projection energy of past and current data; and (ii) an isolated subspace found by maximizing the ratio of new to old energy. The up-projection is learned using a gradient alignment optimization (GAO) scheme. After training each task, closed-form rescaling is performed on rank-1 units to recalibrate the general branch, and then the two branches are merged into the backbone network. Experiments on ImageNet-R/A, CIFAR-100, CUB, DomainNet, and CLIP-based, self-supervised backbones demonstrate that LoDA significantly improves upon the powerful PEFT-CL baseline model.

**Compliance With Llm Reviewing Policy:**

Affirmed.

**Final Justification:**

The rebuttal addressed my concerns.

**Key Questions For Authors:**

The article's experimental results are reliable, its methodology is clear, and it's easy to follow; I tend to approve of it. Of course, some supplementary explanations could better highlight the limitations of the proposed method and point to future research directions. Points that might need further discussion are in the Weakness section.

**Limitations:**

Yes

**Strengths And Weaknesses:**

Strengths:
1. This work, through research and analysis of LoRa, identifies that the loss of LoRa update is gated by the projection energy of the row space of the task features projected downwards, and designs a subspace decomposition design that fits continual learning, thus proposing new insights.
2. The comparative experiments are well-designed, including challenging datasets and state-of-the-art comparative methods, and the appendix includes experiments based on CLIP and a self-supervised backbone. The comprehensive experiments, including ablation and several validation experiments, are well-placed and clearly address any design concerns in advance.
3. The charts are clear and the ideas are easy to follow.

Weakness:
1. One obvious concern is that, although the method can achieve good results without feature replay, the second-moment statistics on which the method relies are, in a sense, contact with the original data, albeit at the cost of low overhead. Of course, completely detaching from prior information is very difficult; this is merely a limitation that needs explanation, not a serious problem.
2. Of course, there are also some possible underlying assumptions, such as these “null bases” of old tasks can remain nearly inactive for new tasks under correlated tasks or S^{1:t-1} is full rank, which are all relatively reasonable.

---

> ### Author Rebuttal · Authors · 2026-03-31
>
> **Q1: On the replay-free claim.**
>
> A: Thanks for your valuable feedback.
>
> We agree that LoDA is not memory-free, since it retains second-moment statistics $S_{1:t-1}\in\mathbb{R}^{D\times D}$ from past tasks.
>
> **Our claim is replay-free.** Compared with replay-based methods, LoDA differs in three ways: (1) **Form of the stored information.** LoDA stores a coupled second-order summary that can be viewed as a **an implicit non-trainable buffer**, rather than explicit replayable sample- or class-level semantic units (e.g.,exemplars or prototypes), thus **does not raise privacy concerns**.(2) **Scalability.** LoDA keeps a fixed-size $D\times D$ matrix per layer, so its storage does not grow with the number of classes or tasks. In contrast, replay-based methods typically require memory that scales with task or class numbers. For example, prototype-based methods such as SSITA require $O(KD^2)$ storage, which grows with the class number $K$. (3) **No replay-based training overhead.** The stored matrix $S$ is a non-trainable buffer used only for subspace construction and closed-form recalibration, rather than being replayed to optimize the backbone or classifier.
>
> We provide a storage comparison between LoDA and existing CL methods in Table 1. Here, $D$ is the feature dimension. $L$ is the number of ViT layers, and $K$ is the class number. For ViT-B/16 on 10S-ImageNetR(INR), we have $D=768$, $L=12$ and $K=200$. Table 1 shows that, compared to representative replay-based SOTAs (SSIAT & SLCA), LoDA achieves a favorable trade-off between storage efficiency and performances.
>
> Table1.
> |Method|Memory-free|Replay-free|Storage Grow with Task Number|Storage Complexity|Storage on 10S-INR(MB)|Acc on 10S-INR|
> |-|-|-|-|-|-|-
> SeqLoRA|Yes|Yes|No|0|0.0|72.09
> InfLoRA|No|Yes|No|$O(LD^2)$|11.3|74.75
> VPT-NSP|No|Yes|No|$O(LD^2)$|27.0|78.88
> SLCA|No|No|Yes|$O(KD^2)$|450.6|79.35
> SSIAT|No|No|Yes|$O(KD^2)$|450.6|79.38
> LoDA|No|Yes|No|$O(LD^2)$|27.0|81.93
>
> **Q2: Assumptions on null-space inactivity.**
>
> A: Thanks for the insightful comment.
>
> This assumption is reasonable in class incremental learning. Although tasks are label-disjoint, they share the same pre-trained feature space and always share common visual patterns. Therefore, **the feature statistics of different tasks are often highly aligned.** To verify this, we measure **the cosine similarities $cos(S_{1:t-1},S_t)$ between old and new task statistics across ViT layers on 10S-ImageNetR (Table 2) and 10S-CIFAR100 (Table 3).** The similarities are consistently high (>0.9), indicating strong statistical alignment between old and new tasks.
>
> Therefore, methods relying on the null-space bases $U_{null}$  of old data tend to remove directions important to the new task, **making the null-space bases nearly inactive for both old and new tasks**. We have empirically validated this phenomenon on 10S-ImageNetA in the paper (Fig.3). Here, we further report the averaged relative new-to-old energy $r(u)$ (defined in Eq.5 in our manuscript) on 10S-ImageNetR and 10S-CIFAR100. The null-space bases found by prior methods (Adam-NSCL and InfLoRA) yield $r(U_{null})\approx1.0$, indicating that $U_{null}$ activated to a similar extent by old and new tasks. This suggests that **the estimated null-space bases are not truly effective and specific for the new task**.
>
> Table 2.
> Layer-Idx|1-3|4-6|7-9|10-12|
> |-|-|-|-|-|
> $cos(S_{1:t-1},S_t)$|0.994|0.992|0.988|0.967
> $r(U_{null})$ (Adam-NSCL)|0.998|1.003|1.135|1.214
> $r(U_{null})$ (InfLoRA)|1.006|1.091|1.245|1.335
> $r(U_I)$ (LoDA)|1.528|1.618|1.917|3.372
>
> Table 3.
> Layer-Idx|1-3|4-6|7-9|10-12|
> |-|-|-|-|-|
> $cos(S_{1:t-1},S_t)$|0.992|0.991|0.986|0.949
> $r(U_{null})$ (Adam-NSCL)|1.120|1.185|1.203|1.271
> $r(U_{null})$ (InfLoRA)|1.186|1.212|1.375|1.549
> $r(U_I)$ (LoDA)|1.698|1.714|2.574|4.526
>
> **Q3: Assumption on full-rank.**
>
> A: Thanks for pointing this out. We note that $S=X^\top X$ is typically full rank in practice on standard real-world datasets. For ViT-B/16, $X\in\mathbb{R}^{(N\times N_p)\times 768}$, where each image contributes $N_p=197$ patch tokens. Even with only 16-shot data, each class already gives $16\times 197=3152$ tokens for a 768-dimensional feature space. Although some work reasonably views token features as approximately low-rank by ignoring small singular values, their variations across spatial locations and images still make $S=X^\top X$ **full-rank in practice without affecting our isolated solver**. As shown by the 16-shot ImageNetR (13% of the full data) ablation in Table 4, the decomposition remains effective in the low-data regime. Moreover, the isolated-subspace solver can be implemented with a small ridge term, i.e., $S\leftarrow S+\epsilon I$, to ensure numerical robustness in ill-conditioned cases.
>
> Table 4.
> ||10S-ImageNetR(16-shot)|20S-ImageNetR(16-shot)|
> |-|-|-|
> SeqLoRA|64.7|61.5
> w/o $LoRA_G$|68.2|67.5
> w/o $LoRA_I$|68.6|66.4
> LoDA|70.2|69.8
>
> We will add the above analysis to our final version.

---

> > ### Author Rebuttal · Reviewer_PEpP · 2026-04-02
> >
> > The rebuttal addressed my concerns. I'd like to raise my score.

---

### Decision · Program_Chairs · 2026-04-30

**Decision:**

Accept (regular)

**Comment:**

While reviewer opinions varied, the rebuttal process clarified the paper’s contributions. The authors addressed critiques regarding theoretical depth by providing rigorous mathematical bounds and a detailed 'Conflict Ratio' analysis. On the empirical side, the work's scope was broadened through new evaluations on TIL and DIL benchmarks against very recent baselines. The AC recommends acceptance, contingent on the inclusion of these rebuttal discussions in the camera-ready version.